# uniINF: Best-of-Both-Worlds Algorithm for Parameter-Free Heavy-Tailed MABs

**Yu Chen** *
IIIS, Tsinghua University
`chenyu23@mails.tsinghua.edu.cn`

**Jiatai Huang** *
Beijing ZhuoShi Capital Co., Ltd., China
`huangjiatai@zsquant.com`

**Yan Dai** *†
ORC, Massachusetts Institute of Technology
`yandai20@mit.edu`

**Longbo Huang** ‡
IIIS, Tsinghua University
`longbohuang@tsinghua.edu.cn`

## Abstract

In this paper, we present a novel algorithm, `uniINF`, for the Heavy-Tailed Multi-Armed Bandits (HTMAB) problem, demonstrating robustness and adaptability in both stochastic and adversarial environments. Unlike the stochastic MAB setting where loss distributions are stationary with time, our study extends to the adversarial setup, where losses are generated from heavy-tailed distributions that depend on both arms and time. Our novel algorithm `uniINF` enjoys the so-called Best-of-Both-Worlds (BoBW) property, performing optimally in both stochastic and adversarial environments *without* knowing the exact environment type. Moreover, our algorithm also possesses a Parameter-Free feature, *i.e.*, it operates *without* the need of knowing the heavy-tail parameters $(\sigma, \alpha)$ a-priori. To be precise, `uniINF` ensures nearly-optimal regret in both stochastic and adversarial environments, matching the corresponding lower bounds when $(\sigma, \alpha)$ is known (up to logarithmic factors). To our knowledge, `uniINF` is the first parameter-free algorithm to achieve the BoBW property for the heavy-tailed MAB problem. Technically, we develop innovative techniques to achieve BoBW guarantees for Parameter-Free HTMABs, including a refined analysis for the dynamics of log-barrier, an auto-balancing learning rate scheduling scheme, an adaptive skipping-clipping loss tuning technique, and a stopping-time analysis for logarithmic regret.

## 1 Introduction

Multi-Armed Bandits (MAB) problem serves as a solid theoretical formulation for addressing the exploration-exploitation trade-off inherent in online learning. Existing research in this area often assumes sub-Gaussian loss (or reward) distributions (Lattimore & Szepesvári, 2020) or even bounded ones (Auer et al., 2002). However, recent empirical evidences revealed that *Heavy-Tailed* (HT) distributions appear frequently in realistic tasks such as network routing (Liebeherr et al., 2012), algorithm portfolio selection (Gagliolo & Schmidhuber, 2011), and online deep learning (Zhang et al., 2020). Such observations underscore the importance of developing MAB solutions that are robust to heavy-tailed distributions.

In this paper, we consider the Heavy-Tailed Multi-Armed Bandits (HTMAB) proposed by Bubeck et al. (2013). In this scenario, the loss distributions associated with each arm do not allow bounded variances but instead have their $\alpha$-th moment bounded by some constant $\sigma^\alpha$, where $\alpha \in (1, 2]$ and $\sigma \geq 0$ are predetermined constants. Mathematically, we assume $\mathbb{E}[|\ell_i|^\alpha] \leq \sigma^\alpha$ for every arm $i$. Although numerous existing HTMAB algorithms operated under the assumption that the parameters $\sigma$ and $\alpha$ are known (Bubeck et al., 2013; Yu et al., 2018; Wei & Srivastava, 2020), real-world applications often present limited knowledge of the true environmental parameters. Thus we propose to explore the scenario where the algorithm lacks any prior information about the parameters. This setup, a variant of classical HTMABs, is referred to as the *Parameter-Free* HTMAB problem.

---

*The first three authors contributed equally to this paper.

†Part of the work was done when Yan Dai was at IIIS, Tsinghua University.

‡Corresponding author.

In addition to whether $\sigma$ and $\alpha$ are known, there is yet another separation that distinguishes our setup from the classical one. In bandit learning literature, there are typically two types of environments, stochastic ones and adversarial ones. In the former type, the loss distributions are always stationary, *i.e.*, they depend solely on the arm and not on time. Conversely, in adversarial environments, the losses can change arbitrarily, as-if manipulated by an adversary aiming to fail our algorithm.

When the environments can be potentially adversarial, a desirable property for bandit algorithms is called *Best-of-Both-Worlds* (BoBW), as proposed by Bubeck & Slivkins (2012). A BoBW algorithm behaves well in both stochastic and adversarial environments without knowing whether stochasticity is satisfied. More specifically, while ensuring near-optimal regret in adversarial environments, the algorithmic performance in a stochastic environment should automatically be boosted, ideally matching the optimal performance of those algorithms specially crafted for stochastic environments.

In practical applications of machine learning and decision-making, acquiring prior knowledge about the environment is often a considerable challenge (Talaei Khoei & Kaabouch, 2023). This is not only regarding the distribution properties of rewards or losses, which may not conform to idealized assumptions such as sub-Gaussian or bounded behaviors, but also about the stochastic or adversarial settings the agent palying in. Such environments necessitate robust solutions that can adapt without prior distributional knowledge. Therefore, the development of a BoBW algorithm that operates effectively without this prior information – termed a parameter-free HTMAB algorithm – is not just a theoretical interest but a practical necessity.

Various previous work has made strides in enhancing the robustness and applicability of HTMAB algorithms. For instance, Huang et al. (2022) pioneered the development of the first BoBW algorithm for the HTMAB problem, albeit requiring prior knowledge of the heavy-tail parameters $(\alpha, \sigma)$ to achieve near-optimal regret guarantees. Genalti et al. (2024) proposed an Upper Confidence Bound (UCB) based parameter-free HTMAB algorithm, specifically designed for stochastic environments. Both algorithms' regret adheres to the instance-dependent and instance-independent lower bounds established by Bubeck et al. (2013). Though excited progress has been made, the following important question still stays open, which further eliminates the need of prior knowledge about the environment:

*Can we design a BoBW algorithm for the Parameter-Free HTMAB problem?*

Addressing parameter-free HTMABs in both adversarial and stochastic environments presents notable difficulties: *i)* although UCB-type algorithms well estimate the underlying loss distribution and provide both optimal instance-dependent and independent regret guarantees, they are incapable of adversarial environments where loss distributions are time-dependent; *ii)* heavy-tailed losses can be potentially very negative, which makes many famous algorithmic frameworks, including Follow-the-Regularized-Leader (FTRL), Online Mirror Descent (OMD), or Follow-the-Perturbed-Leader (FTPL) fall short unless meticulously designed; and *iii)* while it was shown that FTRL with $\beta$-Tsallis entropy regularizer can enjoy best-of-both-worlds guarantees, attaining optimal regret requires an exact match between $\beta$ and $1/\alpha$ — which is impossible without knowing $\alpha$ in advance.

**Our contribution.** In this paper, we answer this open question affirmatively by designing a single algorithm `uniINF` that enjoys both Parameter-Free and BoBW properties — that is, it *i)* does not require any prior knowledge of the environment, *e.g.*, $\alpha$ or $\sigma$, and *ii)* its performance when deployed in an adversarial environment nearly matches the universal instance-independent lower bound given by Bubeck et al. (2013), and it attains the instance-dependent lower bound in stochastic environments as well. For more details, we summarize the advantages of our algorithm in Table 1. Our research directly contributes to enhancing the robustness and applicability of bandit algorithms in a variety of unpredictable and non-ideal conditions. Our main contributions are three-fold:

- We develop a novel BoBW algorithm `uniINF` (see Algorithm 1) for the Parameter-Free HTMAB problem. Without any prior knowledge about the heavy-tail shape-and-scale parameters $(\alpha, \sigma)$, `uniINF` can achieve nearly optimal regret upper bound automatically under both adversarial and stochastic environments (see Table 1 or Theorem 3 for more details).
- We contribute several innovative algorithmic components in designing the algorithm `uniINF`, including a refined analysis for Follow-the-Regularized-Leader (FTRL) with log-barrier regularizers (refined log-barrier analysis in short; see Section 4.1), an auto-balancing learning rate scheduling scheme (see Section 4.2), and an adaptive skipping-clipping loss tuning technique (see Section 4.3).
- To derive the desired BoBW property, we develop novel analytical techniques as well. These include a refined approach to control the Bregman divergence term via calculating partial derivatives and

Table 1: Overview of Related Works For Heavy-Tailed MABs

| Algorithm [a] | $\alpha$-Free? | $\sigma$-Free? | Env. | Regret | Opt? |
|---|---|---|---|---|---|
| Lower Bound (Bubeck et al., 2013) | — | — | — | $\Omega\left(\sum_{i\neq i^*}(\frac{\sigma^\alpha}{\Delta_i})^{\frac{1}{\alpha-1}}\log T\right)$ | — |
| | | | | $\Omega\left(\sigma K^{1-1/\alpha}T^{1/\alpha}\right)$ | — |
| RobustUCB (Bubeck et al., 2013) | ✗ | ✗ | **Only Stoc.** | $\mathcal{O}\left(\sum_{i\neq i^*}(\frac{\sigma^\alpha}{\Delta_i})^{\frac{1}{\alpha-1}}\log T\right)$ | ✓ |
| | | | | $\widetilde{\mathcal{O}}\left(\sigma K^{1-1/\alpha}T^{1/\alpha}\right)$ | ✓ |
| Robust MOSS (Wei & Srivastava, 2020) | ✗ | ✗ | **Only Stoc.** | $\mathcal{O}\left(\sum_{i\neq i^*}(\frac{\sigma^\alpha}{\Delta_i})^{\frac{1}{\alpha-1}}\log(\frac{T}{K}(\frac{\sigma^\alpha}{\Delta_i^\alpha})^{\frac{1}{\alpha-1}})\right)$ | ✓ [b] |
| | | | | $\mathcal{O}\left(\sigma K^{1-1/\alpha}T^{1/\alpha}\right)$ | ✓ |
| APE$^2$ (Lee et al., 2020b) | ✗ | ✓ | **Only Stoc.** | $\mathcal{O}\left(e^\sigma + \sum_{i\neq i^*}(\frac{1}{\Delta_i})^{\frac{1}{\alpha-1}}(T\Delta_i^{\frac{\alpha}{\alpha-1}}\log K)^{\frac{\alpha}{(\alpha-1)\log K}}\right)$ | ✗ |
| | | | | $\widetilde{\mathcal{O}}\left(\exp(\sigma^{1/\alpha})K^{1-1/\alpha}T^{1/\alpha}\right)$ | ✗ |
| HTINF (Huang et al., 2022) | ✗ | ✗ | Stoc. | $\mathcal{O}\left(\sum_{i\neq i^*}(\frac{\sigma^\alpha}{\Delta_i})^{\frac{1}{\alpha-1}}\log T\right)$ | ✓ |
| | | | Adv. | $\mathcal{O}\left(\sigma K^{1-1/\alpha}T^{1/\alpha}\right)$ | ✓ |
| OptHTINF (Huang et al., 2022) | ✓ | ✓ | Stoc. | $\mathcal{O}\left(\sum_{i\neq i^*}(\frac{\sigma^{2\alpha}}{\Delta_i^{3-\alpha}})^{\frac{1}{\alpha-1}}\log T\right)$ | ✗ |
| | | | Adv. | $\mathcal{O}\left(\sigma^\alpha K^{\frac{\alpha-1}{2}}T^{\frac{3-\alpha}{2}}\right)$ | ✗ |
| AdaTINF (Huang et al., 2022) | ✓ | ✓ | **Only Adv.** | $\mathcal{O}\left(\sigma K^{1-1/\alpha}T^{1/\alpha}\right)$ | ✓ |
| AdaR-UCB (Genalti et al., 2024) | ✓ | ✓ | **Only Stoc.** | $\mathcal{O}\left(\sum_{i\neq i^*}(\frac{\sigma^\alpha}{\Delta_i})^{\frac{1}{\alpha-1}}\log T\right)$ | ✓ |
| | | | | $\widetilde{\mathcal{O}}\left(\sigma K^{1-1/\alpha}T^{1/\alpha}\right)$ | ✓ |
| uniINF **(Ours)** | ✓ | ✓ | Stoc. | $\mathcal{O}\left(K(\frac{\sigma^\alpha}{\Delta_{\min}})^{\frac{1}{\alpha-1}}\log T \cdot \log\frac{\sigma^\alpha}{\Delta_{\min}}\right)$ | ✓ [c] |
| | | | Adv. | $\widetilde{\mathcal{O}}\left(\sigma K^{1-1/\alpha}T^{1/\alpha}\right)$ | ✓ |

[a] **$\alpha$-Free?** and **$\sigma$-Free?** denotes whether the algorithm is parameter-free *w.r.t.* $\alpha$ and $\sigma$, respectively. **Env.** includes the environments that the algorithm can work; if one algorithm can work in ***both*** stochastic and adversarial environments, then we mark this column by green. **Regret** describes the algorithmic guarantees, usually (if applicable) instance-dependent ones above instance-independent ones. **Opt?** means whether the algorithm matches the instance-dependent lower bound by Bubeck et al. (2013) *up to constant factors*, or the instance-independent lower bound *up to logarithmic factors*.

[b] Up to $\log(\sigma^\alpha)$ and $\log(1/\Delta_i^\alpha)$ factors.

[c] Up to $\log(\sigma^\alpha)$ and $\log(1/\Delta_{\min})$ factors when all $\Delta_i$'s are similar to the dominant sub-optimal gap $\Delta_{\min}$.

invoking the intermediate value theorem (see Section 5.2) and a stopping-time analysis for achieving $\mathcal{O}(\log T)$ regret in stochastic environments (see Section 5.4).

## 2 RELATED WORK

Due to space limitations, we defer the detailed comparison with the related works into Appendix A and only discuss a few most related ones here.

**Heavy-Tailed Multi-Armed Bandits.** HTMABs were introduced by Bubeck et al. (2013), who gave both instance-dependent and independent lower and upper bounds under stochastic assumptions. Various efforts have been devoted in this area since then. To exemplify, Wei & Srivastava (2020) improve the instance-independent upper bound; Yu et al. (2018) developed a pure exploration algorithm for HTMABs; Medina & Yang (2016); Kang & Kim (2023); Xue et al. (2024) considered the linear HTMAB problem; and Dorn et al. (2024) investigated the symmetric reward distribution.

**Best-of-Both-Worlds Algorithms.** Bubeck & Slivkins (2012) pioneered the study of BoBW bandit algorithms, and was followed by various improvements including EXP3-based approaches (Seldin & Slivkins, 2014), FTPL-based approaches (Honda et al., 2023), and FTRL-based approaches (Wei & Luo, 2018; Zimmert & Seldin, 2019). When the loss distribution can be heavy-tailed, Huang et al. (2022) achieves best-of-both-worlds property under the known-$(\alpha, \sigma)$ assumption.

**Parameter-Free HTMABs.** Another line of HTMABs' research aimed at getting rid of the prior knowledge of $\alpha$ or $\sigma$, which we call Parameter-Free HTMABs. Along this line, Kagrecha et al. (2019) presented the GSR method to identify the optimal arm in HTMAB without any prior knowledge. In terms of regret minimization, Lee et al. (2020b) and Lee & Lim (2022) considered the case when $\sigma$ is

unknown. Genalti et al. (2024) were the first to achieve the parameter-free property while maintaining near-optimal regret. However, all these algorithms fail in adversarial environments.

# 3 PRELIMINARIES: HEAVY-TAILED MULTI-ARMED BANDITS

**Notations.** For an integer $n \geq 1$, $[n]$ denotes the set $\{1, 2, \ldots, n\}$. For a finite set $\mathcal{X}$, $\Delta(\mathcal{X})$ denotes the set of probability distributions over $\mathcal{X}$, often also referred to as the simplex over $\mathcal{X}$. We also use $\Delta^{[K]} := \Delta([K])$ to denote the simplex over $[K]$. We use $\mathcal{O}$ to hide all constant factors, and use $\tilde{\mathcal{O}}$ to additionally suppress all logarithmic factors. We usually use bold letters $\boldsymbol{x}$ to denote a vector, while $x_i$ denotes an entry of the vector. Unless mentioned explicitly, $\log(x)$ denotes the natural logarithm of $x$. Throughout the text, we will use $\{\mathcal{F}_t\}_{t=0}^T$ to denote the natural filtration, *i.e.*, $\mathcal{F}_t$ represents the $\sigma$-algebra generated by all random observations made during the first $t$ time-slots.

Multi-Armed Bandits (MAB) is an interactive game between a player and an environment that lasts for a finite number of $T > 0$ rounds. In each round $t \in [T]$, the player can choose an action from $K > 0$ arms, denoted by $i_t \in [K]$. Meanwhile, a $K$-dimensional loss vector $\boldsymbol{\ell}_t \in \mathbb{R}^K$ is generated by the environment from distribution $\boldsymbol{\nu}_t$, simultaneously without observing $i_t$. The player then suffers a loss of $\ell_{t,i_t}$ and observe this loss (but not the whole loss vector $\boldsymbol{\ell}_t$). The player's objective is to minimize the expected total loss, or equivalently, minimize the following (pseudo-)regret:

$$\mathcal{R}_T := \max_{i \in [K]} \mathbb{E}\left[\sum_{t=1}^T \ell_{t,i_t} - \sum_{t=1}^T \ell_{t,i}\right], \tag{1}$$

where the expectation is taken *w.r.t.* the randomness when the player decides the action $i_t$ and the environment generates the loss $\boldsymbol{\ell}_t$. We use $i^* = \operatorname{argmin}_i \mathbb{E}\left[\sum_{t=1}^T \ell_{t,i}\right]$ to denote the optimal arm.

In Heavy-Tailed MABs (HTMAB), for every $t \in [T]$ and $i \in [K]$, the loss is independently sampled from some heavy-tailed distribution $\nu_{t,i}$ in the sense that $\mathbb{E}_{\ell \sim \nu_{t,i}}[|\ell|^\alpha] \leq \sigma^\alpha$, where $\alpha \in (1, 2]$ and $\sigma \geq 0$ are some pre-determined but *unknown* constants. A Best-of-Both-Worlds (BoBW) algorithm is one that behaves well in both stochastic and adversarial environments (Bubeck & Slivkins, 2012), where stochastic environments are those with time-homogeneous $\{\boldsymbol{\nu}_t\}_{t \in [T]}$ (*i.e.*, $\nu_{t,i} = \nu_{1,i}$ for every $t \in [T]$ and $i \in [K]$) and adversarial environments are those where $\nu_{t,i}$'s can depend on both $t$ and $i$. However, we do not allow the loss distributions to depend on the player's previous actions.[3]

Before concluding this section, we make the following important assumption which is also considered in Huang et al. (2022) and Genalti et al. (2024). Notice that Assumption 1 is strictly weaker than the common "non-negative losses" assumption in the MABs literature (Auer et al., 2002; Lee et al., 2020a; Jin et al., 2023), which only needs the truncated non-negativity for the optimal arm. We make a detailed discussion on this assumption in Appendix B.

**Assumption 1** (Truncated Non-Negative Loss (Huang et al., 2022, Assumption 3.6))**.** *There exits an optimal arm $i^* \in [K]$ such that $\ell_{t,i^*}$ is truncated non-negative for all $t \in [T]$, where a random variable $X$ is called truncated non-negative if $\mathbb{E}[X \cdot \mathbb{1}[|X| > M]] \geq 0$ for any $M \geq 0$.*

Additionally, we make the following assumption for stochastic cases. Assumption 2 is common for algorithms utilizing self-bounding analyses, especially those with BoBW properties (Gaillard et al., 2014; Luo & Schapire, 2015; Wei & Luo, 2018; Zimmert & Seldin, 2019; Ito et al., 2022).[4]

**Assumption 2** (Unique Best Arm)**.** *In stochastic setups, if we denote the mean of distribution $\nu_{1,i}$ as $\mu_i := \mathbb{E}_{\ell \sim \nu_{1,i}}[\ell]$ for all $i \in [K]$, then there exists a unique best arm $i^* \in [K]$ such that $\Delta_i := \mu_i - \mu_{i^*} > 0$ for all $i \neq i^*$. That is, the minimum gap $\Delta_{\min} := \min_{i \neq i^*} \Delta_i$ is positive.*

# 4 THE BOBW HTMAB ALGORITHM uniINF

In this section, we introduce our novel algorithm uniINF (Algorithm 1) for parameter-free HTMABs achieving BoBW. To tackle the adversarial environment, we adopt the famous Follow-the-Regularized-Leader (FTRL) framework instead the statistics-based approach. Moreover, we utilize the log-barrier

---

[3]Called oblivious adversary model (Bubeck & Slivkins, 2012; Wei & Luo, 2018; Zimmert & Seldin, 2019).

[4]While Assumption 2 is commonly adopted in prior work, it is worth noting that some Tsallis-INF algorithms do not require this assumption as analyzed in Ito (2021a) and further extended in Jin et al. (2023).

---

**Algorithm 1** `uniINF`: the universal INF-type algorithm for Parameter-Free HTMAB

---

1: Initialize the learning rate $S_1 \leftarrow 4$.
2: **for** $t = 1, 2, \ldots, T$ **do**
3: Apply *Follow-the-Regularized-Leader* (FTRL) to calculate the action $\boldsymbol{x}_t \in \Delta^{[K]}$ with the log-barrier regularizer $\Psi_t$ defined in Eq. (2): ▷ Refined log-barrier analysis; see Section 4.1.

$$\boldsymbol{x}_t \leftarrow \operatorname*{argmin}_{\boldsymbol{x} \in \Delta^{[K]}} \left( \sum_{s=1}^{t-1} \left\langle \widetilde{\boldsymbol{\ell}}_s, \boldsymbol{x} \right\rangle + \Psi_t(\boldsymbol{x}) \right), \quad \Psi_t(\boldsymbol{x}) := -S_t \sum_{i=1}^{K} \log x_i \tag{2}$$

4: Sample action $i_t \sim \boldsymbol{x}_t$. Play $i_t$ and observe feedback $\ell_{t,i_t}$.
5: **for** $i = 1, 2, \ldots, K$ **do** ▷ Adaptive skipping-clipping loss tuning; see Section 4.3. Note that only $\ell_{t,i_t}^{\text{skip}}$ and $\ell_{t,i_t}^{\text{clip}}$ (but not the whole $\boldsymbol{\ell}_t^{\text{skip}}$ and $\boldsymbol{\ell}_t^{\text{clip}}$ vectors) are accessible to the player.
6:  Calculate the *action-dependent skipping threshold* for arm $i$ and round $t$

$$C_{t,i} := \frac{S_t}{4(1 - x_{t,i})}, \tag{3}$$

and define a *skipped* version and a *clipped* version of the actual loss $\ell_{t,i}$

$$\ell_{t,i}^{\text{skip}} := \text{Skip}(\ell_{t,i}, C_{t,i}) := \begin{cases} \ell_{t,i} & \text{if } |\ell_{t,i}| < C_{t,i} \\ 0 & \text{otherwise} \end{cases},$$

$$\ell_{t,i}^{\text{clip}} := \text{Clip}(\ell_{t,i}, C_{t,i}) := \begin{cases} C_{t,i} & \text{if } \ell_{t,i} \geq C_{t,i} \\ -C_{t,i} & \text{if } \ell_{t,i} \leq -C_{t,i} \\ \ell_{t,i} & \text{otherwise} \end{cases}.$$

7: Calculate the importance sampling estimate of $\boldsymbol{\ell}_t^{\text{skip}}$, namely $\widetilde{\boldsymbol{\ell}}_t$, where $\widetilde{\ell}_{t,i} = \frac{\ell_{t,i}^{\text{skip}}}{x_{t,i}} \cdot \mathbb{1}[i = i_t], \quad \forall i \in [K]$.
8: Update the learning rate $S_{t+1}$ as ▷ Auto-balancing learning rates; see Section 4.2.

$$S_{t+1}^2 = S_t^2 + (\ell_{t,i_t}^{\text{clip}})^2 \cdot (1 - x_{t,i_t})^2 \cdot (K \log T)^{-1}. \tag{4}$$

---

regularizer to derive the logarithmic regret bound in stochastic setting. In the rest of this section, we introduce the main novel components in `uniINF`, including the refined log-barrier analysis (see Section 4.1), the auto-balancing learning rate scheduling scheme (see Section 4.2), and the adaptive skipping-clipping loss tuning technique (see Section 4.3).

## 4.1 REFINED LOG-BARRIER ANALYSIS

We adopt the *log-barrier* regularizer $\Psi_t(\boldsymbol{x}) := -S_t \sum_{i=1}^{K} \log x_i$ in Eq. (2) where $S_t^{-1}$ is the learning rate in round $t$. While log-barrier regularizers were commonly used in the literature for data-adaptive bounds such as small-loss bounds (Foster et al., 2016), path-length bounds (Wei & Luo, 2018), and second-order bounds (Ito, 2021b), and have also been shown to enable the best-of-both-worlds property in certain settings, this paper introduces novel analysis illustrating that log-barrier regularizers also provide environment-adaptivity for both stochastic and adversarial settings.

Precisely, it is known that log-barrier applied to a loss sequence $\{\boldsymbol{c}_t\}_{t=1}^{T}$ ensures $\sum_{t=1}^{T} \langle \boldsymbol{x}_t - \boldsymbol{y}, \boldsymbol{c}_t \rangle \lesssim \sum_{t=1}^{T} ((S_{t+1} - S_t) K \log T + \text{DIV}_t)$ where $\text{DIV}_t \leq S_t^{-1} \sum_{i=1}^{K} x_{t,i}^2 c_{t,i}^2$ for *non-negative* $\boldsymbol{c}_t$'s (Foster et al., 2016, Lemma 16) and $\text{DIV}_t \leq S_t^{-1} \sum_{i=1}^{K} x_{t,i} c_{t,i}^2$ for *general* $\boldsymbol{c}_t$'s (Dai et al., 2023, Lemma 3.1). In comparison, our Lemmas 4 and 5 focus on the case where $S_t$ is *adequately large* compared to $\|\boldsymbol{c}_t\|_\infty$ and give a refined version of $\text{DIV}_t \leq S_t^{-1} \sum_{i=1}^{K} x_{t,i}^2 (1 - x_{t,i})^2 c_{t,i}^2$, which means

$$\sum_{t=1}^{T} \langle \boldsymbol{x}_t - \boldsymbol{y}, \boldsymbol{c}_t \rangle \lesssim \sum_{t=1}^{T} (S_{t+1} - S_t) K \log T + \sum_{t=1}^{T} S_t^{-1} \sum_{i=1}^{K} x_{t,i}^2 (1 - x_{t,i})^2 c_{t,i}^2. \tag{5}$$

The extra $(1 - x_{t,i})^2$ terms are essential to exclude the optimal arm $i^* \in [K]$ — a nice property that leads to best-of-both-worlds guarantees (Zimmert & Seldin, 2019; Dann et al., 2023b). To give more technical details on why we need this $(1 - x_{t,i})^2$, in Section 5.1, we will decompose the regret into skipping error $\sum_{t=1}^{T}(\ell_{t,i_t} - \ell_{t,i_t}^{\text{skip}})\mathbb{1}[i_t \neq i^*]$ and main regret which is roughly $\sum_{t=1}^{T}\langle \boldsymbol{x}_t - \boldsymbol{y}, \boldsymbol{\ell}_t^{\text{skip}}\rangle$. The skipping errors already include the indicator $\mathbb{1}[i_t \neq i^*]$, so the exclusion of $i^*$ is automatic. However, for the main regret, we must manually introduce some $(1 - x_{t,i})$ to exclude $i^*$ — which means the previous bounds mentioned above do not apply, while our novel Eq. (5) is instead helpful.

### 4.2 AUTO-BALANCING LEARNING RATE SCHEDULING SCHEME

We design the our auto-balancing learning rate $S_t$ in Eq. (4). The idea is to balance a Bregman divergence term $\text{DIV}_t$ and a $\Psi$-shifting term $\text{SHIFT}_t$ that arise in our regret analysis (roughly corresponding to the terms on the RHS of Eq. (5)). They allow the following upper bounds as we will include as Lemmas 5 and 9:

$$\underbrace{\text{DIV}_t \leq \mathcal{O}\left(S_t^{-1}\left(\ell_{t,i_t}^{\text{clip}}\right)^2(1 - x_{t,i_t})^2\right)}_{\text{Bregman Divergence}}, \quad \underbrace{\text{SHIFT}_t \leq \mathcal{O}\left((S_{t+1} - S_t) \cdot K\log T\right)}_{\Psi\text{-Shifting}}. \quad (6)$$

Thus, to make $\text{DIV}_t$ roughly the same as $\text{SHIFT}_t$, it suffices to ensure $(S_{t+1} - S_t)S_t \approx (\ell_{t,i_t}^{\text{clip}})^2(1 - x_{t,i_t})^2 \cdot (K\log T)^{-1}$. Our definition of $S_t$ in Eq. (4) follows since $(S_{t+1} - S_t)S_t \approx S_{t+1}^2 - S_t^2$.

### 4.3 ADAPTIVE SKIPPING-CLIPPING LOSS TUNING TECHNIQUE

For heavy-tailed losses, a unique challenge is that $\mathbb{E}[\ell_{t,i_t}^2]$, the squared incurred loss appearing in the Bregman divergence term (as shown in Eq. (6)), does not allow a straightforward upper bound as we only have $\mathbb{E}[|\ell_{t,i_t}|^\alpha] \leq \sigma^\alpha$. A previous workaround is deploying a *skipping* technique that replaces large loss with $0$ (Huang et al., 2022). This technique forbids a sudden increase in the divergence term and thus eases the balance between $\text{DIV}_t$ and $\text{SHIFT}_t$. However, this skipping is not a free lunch: if we skipped too much, the increase of $S_t$ will be slow, which makes the skipping threshold $C_{t,i}$ grow slowly as well — therefore, we end up skipping even more and incurring tremendous *skipping error*! This eventually stops us from establishing instance-dependent guarantees in stochastic setups.

To address this limitation, we develop an *adaptive skipping-clipping* technique. The idea is to ensure that every loss will influence the learning process — imagine that the green $\ell_{t,i_t}^{\text{clip}}$ in Eq. (4) is replaced by $\ell_{t,i_t}^{\text{skip}}$, then $S_t$ will not change if the current $\ell_{t,i_t}$ is large, which risks the algorithm from doing nothing if happening repeatedly. Instead, our clipping technique induces an adequate reaction upon observing a large loss, which both prevents the learning rate $S_t$ from having a drastic change and ensures the growth of $S_t$ is not too slow. Specifically, from Eq. (4), if we skip the loss $\ell_{t,i_t}$, we must have $S_{t+1} = \Theta(1 + (K\log T)^{-1})S_t$. This is crucial for our stopping-time analysis in Section 5.2.

While clipping is used to tune $S_t$, we use the skipped loss $\boldsymbol{\ell}_t^{\text{skip}}$ (more specifically, its importance-weighted version $\widetilde{\boldsymbol{\ell}}_t$) to decide the action $\boldsymbol{x}_t$ in the FTRL update Eq. (2). Utilizing the truncated non-negativity assumption (Assumption 1), we can exclude one arm when controlling the skipping error, as already seen in Eq. (5). We shall present more details in Section 5.4.

## 5 MAIN RESULTS

The main guarantee of our `uniINF` (Algorithm 1) is presented in Theorem 3 below, which states that `uniINF` achieves both optimal minimax regret for adversarial cases (up to logarithmic factors) and near-optimal instance-dependent regret for stochastic cases.

**Theorem 3** (Main Guarantee). *Under the adversarial environments,* `uniINF` *(Algorithm 1) achieves*

$$\mathcal{R}_T = \widetilde{\mathcal{O}}\left(\sigma K^{1-1/\alpha}T^{1/\alpha}\right).$$

*Moreover, for the stochastic environments,* `uniINF` *(Algorithm 1) guarantees*

$$\mathcal{R}_T = \mathcal{O}\left(K\left(\frac{\sigma^\alpha}{\Delta_{\min}}\right)^{\frac{1}{\alpha-1}}\log T \cdot \log\frac{\sigma^\alpha}{\Delta_{\min}}\right).$$

The formal proof of this theorem is provided in Appendix F. As shown in Table 1, our `uniINF` automatically achieves nearly optimal instance-dependent and instance-independent regret guarantees in stochastic and adversarial environments, respectively. Specifically, under stochastic settings, `uniINF` achieves the regret upper bound $\mathcal{O}\left(K\left(\sigma^\alpha/\Delta_{\min}\right)^{1/\alpha-1}\log T \cdot \log \sigma^\alpha/\Delta_{\min}\right)$. Compared to the instance-dependent lower bound $\Omega\left(\sum_{i\neq i^*}(\sigma^\alpha/\Delta_i)^{1/\alpha-1}\log T\right)$ given by Bubeck et al. (2013), our result matches the lower bound up to logarithmic factors $\log \sigma^\alpha/\Delta_{\min}$ which is independent of $T$ when all $\Delta_i$'s are similar to the dominant sub-optimal gap $\Delta_{\min}$. For adversarial environments, our algorithm achieves an $\widetilde{\mathcal{O}}\left(\sigma K^{1-1/\alpha}T^{1/\alpha}\right)$ regret, which matches the instance-independent lower bound $\Omega\left(\sigma K^{1-1/\alpha}T^{1/\alpha}\right)$ given in Bubeck et al. (2013) up to logarithmic terms. Therefore, the regret guarantees of `uniINF` are nearly-optimal in both stochastic and adversarial environments.

## 5.1 Regret Decomposition

In Sections 5.1 to 5.4, we sketch the proof of our BoBW result in Theorem 3. To begin with, we decompose the regret $\mathcal{R}_T$ into a few terms and handle each of them separately. Denoting $\boldsymbol{y}\in\mathbb{R}^K$ as the one-hot vector on the optimal action $i^*\in[K]$, i.e., $y_i := \mathbb{1}[i=i^*]$, we know from Eq. (1) that

$$\mathcal{R}_T = \mathbb{E}\left[\sum_{t=1}^T\langle\boldsymbol{x}_t-\boldsymbol{y},\boldsymbol{\ell}_t\rangle\right].$$

As the log-barrier regularizer $\Psi_t$ is prohibitively large when close to the boundary of $\Delta^{[K]}$, we instead consider the adjusted benchmark $\widetilde{\boldsymbol{y}}$ defined as $\widetilde{y}_i := \begin{cases}\frac{1}{T} & i\neq i^* \\ 1-\frac{K-1}{T} & i=i^*\end{cases}$ and rewrite $\mathcal{R}_T$ as

$$\mathcal{R}_T = \underbrace{\mathbb{E}\left[\sum_{t=1}^T\langle\widetilde{\boldsymbol{y}}-\boldsymbol{y},\boldsymbol{\ell}_t^{\text{skip}}\rangle\right]}_{\text{I. Benchmark Calibration Error}} + \underbrace{\mathbb{E}\left[\sum_{t=1}^T\langle\boldsymbol{x}_t-\widetilde{\boldsymbol{y}},\boldsymbol{\ell}_t^{\text{skip}}\rangle\right]}_{\text{II. Main Regret}} + \underbrace{\mathbb{E}\left[\sum_{t=1}^T\langle\boldsymbol{x}_t-\boldsymbol{y},\boldsymbol{\ell}_t-\boldsymbol{\ell}_t^{\text{skip}}\rangle\right]}_{\text{III. Skipping Error}}. \quad (7)$$

We now go over each term one by one.

**Term I. Benchmark Calibration Error.** As in a typical log-barrier analysis (Wei & Luo, 2018; Ito, 2021b), the Benchmark Calibration Error is not the dominant term. This is because

$$\mathbb{E}\left[\sum_{t=1}^T\langle\widetilde{\boldsymbol{y}}-\boldsymbol{y},\boldsymbol{\ell}_t^{\text{skip}}\rangle\right] \le \sum_{t=1}^T\frac{K-1}{T}\mathbb{E}[|\ell_{t,i_t}^{\text{skip}}|] \le \sum_{t=1}^T\frac{K-1}{T}\mathbb{E}[|\ell_{t,i_t}|] \le \sigma K,$$

which is independent from $T$. Therefore, the key is analyzing the other two terms.

**Term II. Main Regret.** By FTRL regret decomposition (see Lemma 29 in the appendix; it is an extension of the classical FTRL bounds (Lattimore & Szepesvári, 2020, Theorem 28.5)), we have

$$\text{Main Regret} = \mathbb{E}\left[\sum_{t=1}^T\langle\boldsymbol{x}_t-\widetilde{\boldsymbol{y}},\boldsymbol{\ell}_t^{\text{skip}}\rangle\right] = \mathbb{E}\left[\sum_{t=1}^T\langle\boldsymbol{x}_t-\widetilde{\boldsymbol{y}},\widetilde{\boldsymbol{\ell}}_t\rangle\right]$$

$$\le \sum_{t=1}^T\mathbb{E}[D_{\Psi_t}(\boldsymbol{x}_t,\boldsymbol{z}_t)] + \sum_{t=0}^{T-1}\mathbb{E}\left[(\Psi_{t+1}(\widetilde{\boldsymbol{y}})-\Psi_t(\widetilde{\boldsymbol{y}}))-(\Psi_{t+1}(\boldsymbol{x}_{t+1})-\Psi_t(\boldsymbol{x}_{t+1}))\right],$$

where $D_{\Psi_t}(\boldsymbol{y},\boldsymbol{x})=\Psi_t(\boldsymbol{y})-\Psi_t(\boldsymbol{x})-\langle\nabla\Psi_t(\boldsymbol{x}),\boldsymbol{y}-\boldsymbol{x}\rangle$ is the Bregman divergence induced by the $t$-th regularizer $\Psi_t$, and $\boldsymbol{z}_t$ denotes the posterior optimal estimation in episode $t$, namely

$$\boldsymbol{z}_t := \operatorname*{argmin}_{\boldsymbol{z}\in\Delta^{[K]}}\left(\sum_{s=1}^t\langle\widetilde{\boldsymbol{\ell}}_s,\boldsymbol{z}\rangle+\Psi_t(\boldsymbol{z})\right). \quad (8)$$

For simplicity, we use the abbreviation $\text{Div}_t := D_{\Psi_t}(\boldsymbol{x}_t,\boldsymbol{z}_t)$ for the Bregman divergence between $\boldsymbol{x}_t$ and $\boldsymbol{z}_t$ under regularizer $\Psi_t$, and let $\text{Shift}_t := [(\Psi_{t+1}(\widetilde{\boldsymbol{y}})-\Psi_t(\widetilde{\boldsymbol{y}}))-(\Psi_{t+1}(\boldsymbol{x}_{t+1})-\Psi_t(\boldsymbol{x}_{t+1}))]$ be the $\Psi$-shifting term. Then, we can reduce the analysis of main regret to bounding the sum of Bregman divergence term $\mathbb{E}[\text{Div}_t]$ and $\Psi$-shifting term $\mathbb{E}[\text{Shift}_t]$.

**Term III. Skipping Error.** To control the skipping error, we define $\text{SKIPERR}_t := \ell_{t,i_t} - \ell_{t,i_t}^{\text{skip}} = \ell_{t,i_t} \mathbb{1}[|\ell_{t,i_t}| \geq C_{t,i_t}]$ as the loss incurred by the skipping operation at episode $t$. Then we have

$$\langle \boldsymbol{x}_t - \boldsymbol{y}, \boldsymbol{\ell}_t - \boldsymbol{\ell}_t^{\text{skip}} \rangle = \sum_{i \in [K]} (x_{t,i} - y_i) \cdot (\ell_{t,i} - \ell_{t,i}^{\text{skip}})$$

$$\leq \sum_{i \neq i^*} x_{t,i} \cdot \left| \ell_{t,i} - \ell_{t,i}^{\text{skip}} \right| + (x_{t,i^*} - 1) \cdot \left( \ell_{t,i^*} - \ell_{t,i^*}^{\text{skip}} \right)$$

$$= \mathbb{E}\left[ |\text{SKIPERR}_t| \cdot \mathbb{1}[i_t \neq i^*] \mid \mathcal{F}_{t-1} \right] + (x_{t,i^*} - 1) \cdot \left( \ell_{t,i^*} - \ell_{t,i^*}^{\text{skip}} \right).$$

Notice that the factor $(x_{t,i^*} - 1)$ in the second term is negative and $\mathcal{F}_{t-1}$-measurable. Then we have

$$\mathbb{E}\left[ \ell_{t,i^*} - \ell_{t,i^*}^{\text{skip}} \Big| \mathcal{F}_{t-1} \right] = \mathbb{E}\left[ \mathbb{1}[|\ell_{t,i^*}| \geq C_{t,i^*}] \cdot \ell_{t,i^*} \right] \geq 0,$$

where the inequality is due to the truncated non-negative assumption (Assumption 1) of the optimal arm $i^*$. Therefore, we have $\mathbb{E}[(x_{t,i^*} - 1) \cdot (\ell_{t,i^*} - \ell_{t,i^*}^{\text{skip}}) \mid \mathcal{F}_{t-1}] \leq 0$ and thus

$$\mathbb{E}[\langle \boldsymbol{x}_t - \boldsymbol{y}, \boldsymbol{\ell}_t - \boldsymbol{\ell}_t^{\text{skip}} \rangle \mid \mathcal{F}_{t-1}] \leq \mathbb{E}[|\text{SKIPERR}_t| \cdot \mathbb{1}[i_t \neq i^*] \mid \mathcal{F}_{t-1}], \tag{9}$$

which gives an approach to control the skipping error by the sum of skipping losses $\text{SKIPERR}_t$'s where we pick a sub-optimal arm $i_t \neq i^*$. Formally, we give the following inequality:

$$\text{SKIPPING ERROR} \leq \mathbb{E}\left[ \sum_{t=1}^{T} |\text{SKIPERR}_t| \cdot \mathbb{1}[i_t \neq i^*] \right].$$

To summarize, the regret $\mathcal{R}_T$ decomposes into the sum of Bregman divergence terms $\mathbb{E}[\text{DIV}_t]$, $\Psi$-shifting terms $\mathbb{E}[\text{SHIFT}_t]$, and sub-optimal skipping losses $\mathbb{E}[|\text{SKIPERR}_t| \cdot \mathbb{1}[i_t \neq i^*]]$, namely

$$\mathcal{R}_T \leq \underbrace{\mathbb{E}\left[ \sum_{t=1}^{T} \text{DIV}_t \right]}_{\text{BREGMAN DIVERGENCE TERMS}} + \underbrace{\mathbb{E}\left[ \sum_{t=0}^{T-1} \text{SHIFT}_t \right]}_{\Psi\text{-SHIFTING TERMS}} + \underbrace{\mathbb{E}\left[ \sum_{t=1}^{T} |\text{SKIPERR}_t| \cdot \mathbb{1}[i_t \neq i^*] \right]}_{\text{SUB-OPTIMAL SKIPPING LOSSES}} + \sigma K. \tag{10}$$

In Sections 5.2 to 5.4, we analyze these three key items and introduce our novel analytical techniques for both adversarial and stochastic cases. Specifically, we bound the Bregman divergence terms in Section 5.2 (all formal proofs in Appendix C), the $\Psi$-shifting terms in Section 5.3 (all formal proofs in Appendix D), and the skipping error terms in Section 5.4 (all formal proofs in Appendix E). Afterwards, to get our Theorem 3, putting Theorems 7, 10, and 13 together gives the adversarial guarantee, while the stochastic guarantee follows from a combination of Theorems 8, 11, and 14.

## 5.2 ANALYZING BREGMAN DIVERGENCE TERMS

From the log-barrier regularizer defined in Eq. (2), we can explicitly write out $\text{DIV}_t$ as

$$\text{DIV}_t = D_{\Psi_t}(\boldsymbol{x}_t, \boldsymbol{z}_t) = S_t \sum_{i \in [K]} \left( -\log \frac{x_{t,i}}{z_{t,i}} + \frac{x_{t,i}}{z_{t,i}} - 1 \right).$$

To simplify notations, we define $w_{t,i} := \frac{x_{t,i}}{z_{t,i}} - 1$, which gives $\text{DIV}_t = S_t \sum_{i=1}^{K}(w_{t,i} - \log(w_{t,i}+1))$. Therefore, one natural idea is to utilize the inequality $x - \log(x + 1) \leq x^2$ for $x \in [-1/2, 1/2]$. To do so, we need to ensure $-1/2 \leq w_{t,i} \leq 1/2$, *i.e.*, $\boldsymbol{z}_t$ is multiplicatively close to $\boldsymbol{x}_t$. If the losses are bounded, this property is achieved in previously papers (Lee et al., 2020a; Jin & Luo, 2020). However, it is non-trivial for heavy-tailed (thus unbounded) losses. By refined analysis on the calculation of $\boldsymbol{x}$ and $\boldsymbol{z}$ by KKT conditions, we develop novel technical tools to depict $\boldsymbol{z}_t$ and provide the following important lemma in HTMABs. We refer readers to the proof of Lemma 15 for more technical details.

**Lemma 4** ($\boldsymbol{z}_t$ is Multiplicatively Close to $\boldsymbol{x}_t$). $\frac{1}{2}x_{t,i} \leq z_{t,i} \leq 2x_{t,i}$ *for every* $t \in [T]$ *and* $i \in [K]$.

Lemma 4 implies $w_{t,i} \in [-1/2, 1/2]$. Hence $\text{DIV}_t = S_t \sum_{t=1}^{K}(w_{t,i} - \log(w_{t,i}+1)) \leq S_t \sum_{t=1}^{K} w_{t,i}^2$. Conditioning on the natural filtration $\mathcal{F}_{t-1}$, $w_{t,i}$ is fully determined by feedback $\widetilde{\boldsymbol{\ell}}_t$ in episode $t$, which allows us to give a precise depiction of $w_{t,i}$ via calculating the partial derivative $\partial w_{t,i}/\partial \widetilde{\ell}_{t,j}$ and invoking the intermediate value theorem. The detailed procedure is included as Lemma 16 in the appendix, which further results in the following lemma on the Bregman divergence term $\text{DIV}_t$.

**Lemma 5** (Upper Bound of $\text{DIV}_t$). *For every $t \in [T]$, we can bound $\text{DIV}_t$ as*

$$\text{DIV}_t = \mathcal{O}\left(S_t^{-1}(\ell_{t,i_t}^{skip})^2(1 - x_{t,i_t})^2\right), \quad \sum_{\tau=1}^{t} \text{DIV}_\tau = \mathcal{O}\left(S_{t+1} \cdot K \log T\right).$$

Compared to previous bounds on $\text{DIV}_t$, Lemma 5 contains an $(1 - x_{t,i_t})^2$ that is crucial for our instance-dependent bound and serves as a main technical contribution, as we sketched in Section 4.1.

**Adversarial Cases.** By definition of $S_t$ in Eq. (4), we can bound $S_{T+1}$ as in the following lemma.

**Lemma 6** (Upper Bound for $S_{T+1}$). *The expectation of $S_{T+1}$ can be bounded by*

$$\mathbb{E}[S_{T+1}] \leq 2 \cdot \sigma (K \log T)^{-1/\alpha} T^{1/\alpha}. \tag{11}$$

Combining this upper-bound on $\mathbb{E}[S_{T+1}]$ with Lemma 5, we can control the expected sum of Bregman divergence terms $\mathbb{E}[\sum_{t=1}^{T} \text{DIV}_t]$ in adversarial environments, as stated in the following theorem.

**Theorem 7** (Adversarial Bounds for Bregman Divergence Terms). *In adversarial environments, the sum of Bregman divergence terms can be bounded by*

$$\mathbb{E}\left[\sum_{t=1}^{T} \text{DIV}_t\right] = \widetilde{\mathcal{O}}\left(\sigma K^{1-1/\alpha} T^{1/\alpha}\right).$$

**Stochastic Cases.** For $\mathcal{O}(\log T)$ bounds, we opt for the first statement of Lemma 5. By definition of $\ell_{t,i_t}^{skip}$, we immediately have $|\ell_{t,i_t}^{skip}| \leq C_{t,i_t} = \mathcal{O}(S_t(1 - x_{t,i_t})^{-1})$. Further using $\mathbb{E}[|\ell_{t,i_t}|^\alpha] \leq \sigma^\alpha$, we have $\mathbb{E}[\text{DIV}_t \mid \mathcal{F}_{t-1}] = \mathcal{O}(S_t^{1-\alpha} \sigma^\alpha (1 - x_{t,i^*}))$ (formalized as Lemma 20 in the appendix).

We can now perform a *stopping-time argument* for the sum of Bregman divergence terms in stochastic cases. Briefly, we pick a fixed constant $M > 0$. The expected sum of $\text{DIV}_t$'s on those $\{t \mid S_t \geq M\}$ is then within $\mathcal{O}(M^{1-\alpha} \cdot T)$ according to Lemma 5. On the other hand, we claim that the sum of $\text{DIV}_t$'s on those $\{t \mid S_t < M\}$ can also be well controlled because $\mathbb{E}[\text{DIV}_t \mid \mathcal{F}_{t-1}] = \mathcal{O}(\mathbb{E}[S_t^{1-\alpha} \sigma^\alpha (1 - x_{t,i^*}) \mid \mathcal{F}_{t-1}])$. The detailed analysis is presented in Appendix C.6, and we summarize it as follows. Therefore, the sum of Bregman divergences in stochastic environments is also well-controlled.

**Theorem 8** (Stochastic Bounds for Bregman Divergence Terms). *In stochastic settings, the sum of Bregman divergence terms can be bounded by*

$$\mathbb{E}\left[\sum_{t=1}^{T} \text{DIV}_t\right] = \mathcal{O}\left(K\sigma^{\frac{\alpha}{\alpha-1}} \Delta_{\min}^{-\frac{1}{\alpha-1}} \log T + \frac{\mathcal{R}_T}{4}\right).$$

### 5.3 ANALYZING $\Psi$-SHIFTING TERMS

Since our choice of $\{S_t\}_{t \in [T]}$ is non-decreasing, the $\Psi$-shifting term $\Psi_{t+1}(\boldsymbol{x}) - \Psi_t(\boldsymbol{x}) \geq 0$ trivially holds for any $\boldsymbol{x} \in \Delta^{[K]}$. We get the following lemma by algebraic manipulations.

**Lemma 9** (Upper Bound of $\text{SHIFT}_t$). *For every $t \geq 0$, we can bound $\text{SHIFT}_t$ as*

$$\text{SHIFT}_t = \mathcal{O}\left(S_t^{-1}(\ell_{t,i_t}^{clip})^2(1 - x_{t,i_t})^2\right), \quad \sum_{\tau=0}^{t} \text{SHIFT}_\tau = \mathcal{O}\left(S_{t+1} \cdot K \log T\right).$$

Similarly, Lemma 9 also contains a useful $(1 - x_{t,i_t})^2$ term — in fact, the two bounds in Lemma 9 are extremely similar to those in Lemma 5 and thus allow analogous analyses. This is actually an expected phenomenon, thanks to our auto-balancing learning rates introduced in Section 4.2.

**Adversarial Cases.** Again, we utilize the $\mathbb{E}[S_{T+1}]$ bound in Lemma 6 and get the following theorem.

**Theorem 10** (Adversarial Bounds for $\Psi$-Shifting Terms). *In adversarial environments, the expectation of the sum of $\Psi$-shifting terms can be bounded by*

$$\mathbb{E}\left[\sum_{t=0}^{T-1} \text{SHIFT}_t\right] \leq \mathbb{E}[S_{T+1} \cdot (K \log T)] = \widetilde{\mathcal{O}}\left(\sigma K^{1-1/\alpha} T^{1/\alpha}\right).$$

**Stochastic Cases.** Still similar to $\text{DIV}_t$'s, we condition on $\mathcal{F}_{t-1}$ and utilize $|\ell_{t,i_t}^{\text{clip}}| \leq C_{t,i_t}$ to get $\mathbb{E}\left[\text{SHIFT}_t \mid \mathcal{F}_{t-1}\right] = \mathcal{O}(S_t^{1-\alpha}\sigma^\alpha(1 - x_{t,i^*}))$ (formalized as Lemma 24 in the appendix). Thus, for stochastic environments, a stopping-time argument similar to that of $\text{DIV}_t$ yields Theorem 11.

**Theorem 11** (Stochastic Bounds for $\Psi$-shifting Terms). *In stochastic environments, the sum of $\Psi$-shifting terms can be bounded by*

$$\mathbb{E}\left[\sum_{t=0}^{T-1}\text{SHIFT}_t\right] = \mathcal{O}\left(K\sigma^{\frac{\alpha}{\alpha-1}}\Delta_{\min}^{-\frac{1}{\alpha-1}}\log T + \frac{\mathcal{R}_T}{4}\right).$$

### 5.4 ANALYZING SUB-OPTIMAL SKIPPING LOSSES

This section controls the sub-optimal skipping losses when we pick a sub-optimal arm $i_t \neq i^*$, *i.e.*,

$$\sum_{t=1}^{T}|\text{SKIPERR}_t \cdot \mathbb{1}[i_t \neq i^*]| = \sum_{t=1}^{T}|\ell_{t,i_t} - \ell_{t,i_t}^{\text{skip}}| \cdot \mathbb{1}[i_t \neq i^*] = \sum_{t=1}^{T}|\ell_{t,i_t}| \cdot \mathbb{1}[|\ell_{t,i_t}| \geq C_{t,i_t}] \cdot \mathbb{1}[i_t \neq i^*].$$

For skipping errors, we need a more dedicated stopping-time analysis in both adversarial and stochastic cases. Different from previous sections, it is now non-trivial to bound the sum of $\text{SKIPERR}_t$'s on those $\{t \mid S_t < M\}$ where $M$ is the stopping-time threshold. However, a key observation is that whenever we encounter a non-zero $\text{SKIPERR}_t$, $S_{t+1}$ will be $\Omega(1)$-times larger than $S_t$. Thus the number of non-zero $\text{SKIPERR}_t$'s before $S_t$ reaching $M$ is small. Equipped with this observation, we derive the following lemma, whose formal proof is included in Appendix E.1.

**Lemma 12** (Stopping-Time Argument for Skipping Losses). *Given a stopping-time threshold $M$, the total skipping loss on those $t$'s with $i_t \neq i^*$ is bounded by*

$$\mathbb{E}\left[\sum_{t=1}^{T}|\text{SKIPERR}_t| \cdot \mathbb{1}[i_t \neq i^*]\right] \leq M\left(\frac{\sigma^\alpha}{M^\alpha}\mathbb{E}\left[\sum_{t=1}^{T}\mathbb{1}[i_t \neq i^*]\right] + 2 \cdot \frac{\log M}{\log\left(1 + \frac{1}{16K\log T}\right)} + 1\right).$$

Equipped with this novel stopping-time analysis, it only remains to pick a proper threshold $M$ for Lemma 12. It turns out that for adversarial and stochastic cases, we have to pick different $M$'s. Specifically, we achieve the following two theorems, whose proofs are in Appendices E.2 and E.3.

**Theorem 13** (Adversarial Bounds for Skipping Losses). *By setting adversarial stopping-time threshold $M^{adv} := \sigma(K\log T)^{-1/\alpha}T^{1/\alpha}$, we have*

$$\mathbb{E}\left[\sum_{t=1}^{T}|\text{SKIPERR}_t| \cdot \mathbb{1}[i_t \neq i^*]\right] = \widetilde{\mathcal{O}}\left(\sigma K^{1-1/\alpha}T^{1/\alpha}\right).$$

**Theorem 14** (Stochastic Bounds for Skipping Losses). *By setting stochastic stopping-time threshold $M^{sto} := 4^{\frac{1}{\alpha-1}}\sigma^{\frac{\alpha}{\alpha-1}}\Delta_{\min}^{-\frac{1}{\alpha-1}}$, we have*

$$\mathbb{E}\left[\sum_{t=1}^{T}|\text{SKIPERR}_t| \cdot \mathbb{1}[i_t \neq i^*]\right] = \mathcal{O}\left(K\log T \cdot \sigma^{\frac{\alpha}{\alpha-1}}\Delta_{\min}^{-\frac{1}{\alpha-1}} \cdot \log\left(\frac{\sigma^\alpha}{\Delta_{\min}}\right) + \frac{\mathcal{R}_T}{4}\right).$$

## 6 CONCLUSION

This paper designs the first algorithm for Parameter-Free HTMABs that enjoys the Best-of-Both-Worlds property. Specifically, our algorithm, `uniINF`, simultaneously achieves near-optimal instance-independent and instance-dependent bounds in adversarial and stochastic environments, respectively. `uniINF` incorporates several innovative algorithmic components, such as *i)* refined log-barrier analysis, *ii)* auto-balancing learning rates, and *iii)* adaptive skipping-clipping loss tuning. Analytically, we also introduce meticulous techniques including *iv)* analyzing Bregman divergence via partial derivatives and intermediate value theorem and *v)* stopping-time analysis for logarithmic regret. We expect many of these techniques to be of independent interest. In terms of limitations, `uniINF` does suffer from some extra logarithmic factors; it's dependency on the gaps $\{\Delta_i^{-1}\}_{i \neq i^*}$ is also improvable when some of the gaps are much smaller than the other. We leave these for future investigation.

## REPRODUCIBILITY STATEMENT

This paper is primarily theoretical in nature, with the main contributions being the development of `uniINF`, the Best-of-Both-Worlds algorithm for Parameter-Free Heavy-Tailed Multi-Armed Bandits, and the introduction of several innovative algorithmic components and analytical tools. To ensure reproducibility, we have provided clear and detailed explanations of our methods, assumptions, and the development process of the algorithm throughout the paper. Complete proofs of our claims and results are included in the appendices. We have made every effort to present our research in a transparent and comprehensive manner to enable other researchers to understand and replicate our results.

## ACKNOWLEDGMENTS

We thank the anonymous reviewers for their detailed reviews, from which we benefit greatly. This work was supported by the National Natural Science Foundation of China Grants 52450016 and 52494974, and in part by the National Science Foundation of China under Grant 623B1015.

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

# Supplementary Materials

# A   ADDITIONAL RELATED WORKS

In this section, we present a detailed version of literature review to include more discussion and related works.

## A.1   MULTI-ARMED BANDITS

The Multi-Armed Bandits (MAB) problem (Thompson, 1933; Slivkins et al., 2019; Lattimore & Szepesvári, 2020; Bubeck et al., 2012) is an important theoretical framework for addressing the exploration-exploitation trade-off inherent in online learning. Various studies discovered different setups for the loss (or reward) signals. For example, most works focus on standard bounded losses where the loss of each arm belongs to a bounded range such as $[0, 1]$ (Auer et al., 2002; Wei & Luo, 2018; Lee et al., 2020a; Jin et al., 2023). Srinivas et al. (2009); Wu et al. (2015); Deng et al. (2022) assumed the losses following Gaussian or sub-Gaussian distribution. Another branch of works called scale-free MABs consider the case when the losses' range is unknown to the agent (Putta & Agrawal, 2022; Chen & Zhang, 2023; 2024; Hadiji & Stoltz, 2023). In this paper, we consider the heavy-tailed loss signals where each arm's loss do not allow bounded variances but instead have their $\alpha$-th moment bounded by some constant $\sigma^\alpha$, which is called the Heavy-Tailed Multi-Armed Bandits (HTMABs).

## A.2   HEAVY-TAILED MABs

HTMABs were introduced by Bubeck et al. (2013), who gave both instance-dependent and instance-independent lower and upper bounds under stochastic assumptions. Various efforts have been devoted in this area since then. To exemplify, Wei & Srivastava (2020) removed a sub-optimal $(\log T)^{1-1/\alpha}$ factor in the instance-independent upper bound; Yu et al. (2018) developed a pure exploration algorithm for HTMABs; Medina & Yang (2016), Kang & Kim (2023), and Xue et al. (2024) considered the linear HTMAB problem; and Dorn et al. (2024) specifically investigated the case where the heavy-tailed reward distributions are presumed to be symmetric. Nevertheless, all these works focused on stochastic environments and required the prior knowledge of heavy-tail parameters $(\alpha, \sigma)$. In contrast, this paper focuses on both-of-both-worlds algorithms in parameter-free HTMABs, which means *i)* the loss distributions can possibly be non-stationary, and *ii)* the true heavy-tail parameters $(\alpha, \sigma)$ remain unknown.

## A.3   BEST-OF-BOTH-WORLDS ALGORITHMS

Bubeck & Slivkins (2012) pioneered the study of Best-of-Both-Worlds bandit algorithms, and was followed by various improvements including EXP3-based approaches (Seldin & Slivkins, 2014; Seldin & Lugosi, 2017), FTPL-based approaches (Honda et al., 2023; Lee et al., 2024), and FTRL-based approaches (Wei & Luo, 2018; Zimmert & Seldin, 2019; Jin et al., 2023). Equipped with data-dependent learning rates, Ito (2021a;b); Tsuchiya et al. (2023); Ito & Takemura (2023); Kong et al. (2023) achieve Best-of-Both-Worlds algorithms in various online learning problems. When the loss distribution can be heavy-tailed, Huang et al. (2022) gave an algorithm HTINF that achieves Best-of-Both-Worlds property under the known-$(\alpha, \sigma)$ assumption. Unfortunately, without access to these true parameters, their alternative algorithm OptHTINF failed to achieve near-optimal regret guarantees in either adversarial or stochastic environments.

## A.4   PARAMETER-FREE HTMABs

Another line of research aimed at getting rid of the prior knowledge of $\alpha$ or $\sigma$, which we call Parameter-Free HTMABs. Along this line, Kagrecha et al. (2019) presented the GSR method to identify the optimal arm in HTMAB without any prior knowledge. In terms of regret minimization, Lee et al. (2020b) and Lee & Lim (2022) considered the case when $\sigma$ is unknown. Genalti et al. (2024) were the first to achieve the parameter-free property while maintaining near-optimal regret. However, all these algorithms fail in adversarial environments – not to mention the best-of-both-worlds property which requires optimality in both stochastic and adversarial environments.

## B   DISCUSSIONS ON ASSUMPTION 1 (TRUNCATED NON-NEGATIVE LOSS)

In this section, we detailedly discuss the existing results on Assumption 1 in HTMABs works. It is important to note that the truncated non-negativity loss assumption is not unique to our study. In fact, it was also used in Huang et al. (2022) and Genalti et al. (2024), and is indeed a relaxation of the more common "non-negative losses" assumption found in the MAB literature (Auer et al., 2002; Lee et al., 2020a; Jin et al., 2023). For a better comparison, we summarize the exisiting results in HTMABs for the relationship between the prior knowledge of heavy-tailed parameters $(\sigma, \alpha)$ and Assumption 1 in the following table.

Table 2: Impact of Assumption 1 on BoBW Guarantees for HTMABs

| Assumptions | Known $(\sigma, \alpha)$ | Unknown $(\sigma, \alpha)$ |
|---|---|---|
| With Assumption 1 | HTINF achieves BoBW (Huang et al., 2022) | uniINF achieves BoBW (**Our Work**) |
| Weaker than Assumption 1? | Inferred by Cheng et al. (2024) (discussed below) | **Open Problem** |
| Without Any Assumption | SAO-HT achieves BoBW (Cheng et al., 2024) | No BoBW possible (Genalti et al., 2024, Theorems 2 & 3) |

The truncated non-negative loss assumption is first introduced by Huang et al. (2022), and their algorithm HTINF achieves nearly optimal automatically in both stochasitc and adversarial environments with Assumption 1 and prior knowledge of heavy-tailed parameters $(\sigma, \alpha)$, as shown in Table 2.

Recently, Cheng et al. (2024) justified that when parameters $(\sigma, \alpha)$ are known, BoBW is achievable without any assumptions (that is, Assumption 1 is redundant when parameters are known). Specifically, the algorithm SAO-HT presented by Cheng et al. (2024) achieves nearly optimal in adversarial and stochastic cases up to some extra logarithm factors. This result implies that Assumption 1 is redundant for optiaml HTMABs when parameters are known.

When it comes to parameter-free setups, Theorems 2 and 3 by Genalti et al. (2024) highlight that achieving optimal worst-case regret guarantees is impossible without any assumptions. Moreover, Genalti et al. (2024) present an efficient algorithm Adar-UCB which achieve nearly minimax optimal on both instance-dependent and instance-independent regret bounds, under the stochastic environments and Assumption 1.

Therefore, we adopt Assumption 1 when developing our BoBW algorithms for parameter-free HTMABs. Our algorithm, uniINF (Algorithm 1), achieves nearly optimal automatically in both stochastic and adversarial settings without knowing the parameters $(\sigma, \alpha)$ a-priori. Our result also demonstrates that Assumption 1 is sufficient for BoBW guarantees in parameter-free HTMABs.

As summarized by Table 2, while Assumption 1 has been validated as sufficient for achieving BoBW guarantees in both known and unknown parameter settings, the exploration for weaker assumptions continues to be an intersting topic for future research.

## C   BREGMAN DIVERGENCE TERMS: OMITTED PROOFS IN SECTION 5.2

In this section, we present the omitted proofs in Section 5.2, the analyses for the Bregman divergence term $\text{DIV}_t = D_{\Psi_t}(\boldsymbol{x}_t, \boldsymbol{z}_t) = \Psi_t(\boldsymbol{x}_t) - \Psi_t(\boldsymbol{z}_t) - \langle \nabla \Psi_t(\boldsymbol{z}_t), \boldsymbol{x}_t - \boldsymbol{z}_t \rangle$ where $\boldsymbol{x}_t, \boldsymbol{z}_t$ are defined as

$$\boldsymbol{x}_t := \underset{\boldsymbol{x} \in \Delta^{[K]}}{\arg\min} \langle \boldsymbol{L}_{t-1}, \boldsymbol{x} \rangle + \Psi_t(\boldsymbol{x}),$$

$$\boldsymbol{z}_t := \underset{\boldsymbol{z} \in \Delta^{[K]}}{\arg\min} \langle \boldsymbol{L}_t, \boldsymbol{z} \rangle + \Psi_t(\boldsymbol{z}),$$

and $\boldsymbol{L}_t$ is defined as the cumulative loss

$$\boldsymbol{L}_t = \sum_{s=1}^{t} \widetilde{\boldsymbol{\ell}}_s, \quad \widetilde{\ell}_{s,i} = \frac{\ell_{t,i}^{\text{skip}}}{x_{t,i}} \cdot \mathbb{1}[i = i_t].$$

By KKT conditions, there exist two unique multipliers $Z_t$ and $\widetilde{Z}_t$ such that

$$x_{t,i} = \frac{S_t}{L_{t-1,i} - Z_t}, \quad z_{t,i} = \frac{S_t}{L_{t,i} - \widetilde{Z}_t}, \quad \forall i \in [K].$$

We highlight that $Z_t$ is fixed conditioning on $\mathcal{F}_{t-1}$, while $\widetilde{Z}_t$ is fully determined by $\widetilde{\ell}_t$.

### C.1 $z_{t,i}$ IS MULTIPLICATIVELY CLOSE TO $x_{t,i}$: PROOF OF LEMMA 4

Lemma 4 investigates the relationship between $x_{t,i}$ and $z_{t,i}$. By previous discussion, the key of the proof is carefully studied the multipliers $Z_t$ and $\widetilde{Z}_t$.

**Lemma 15** (Restatement of Lemma 4). *For every $t \in [T]$ and $i \in [K]$, we have*

$$\frac{1}{2}x_{t,i} \le z_{t,i} \le 2x_{t,i}.$$

*Proof.* **Positive Loss.** Below we first consider $\widetilde{\ell}_{t,i_t} > 0$, the other case $\widetilde{\ell}_{t,i_t} < 0$ can be verified similarly, and the $\widetilde{\ell}_{t,i_t} = 0$ case is trivial. We prove this lemma for left and right sides, respectively.

**The left side $\frac{1}{2}x_{t,i} \le z_{t,i}$:** According to the KKT conditions, given $S_t$ and $L_{t-1}$ ($L_t$), the goal of computing $x_t$ ($z_t$) can be reduced to find a scalar multiplier $Z_t$ ($\widetilde{Z}_t$), such that we have

$$\sum_{i=1}^{K} x_{t,i}(Z_t) = 1$$

and

$$\sum_{i=1}^{K} z_{t,i}(\widetilde{Z}_t) = 1$$

where

$$x_{t,i}(\lambda) = \frac{S_t}{L_{t-1,i} - \lambda}, \quad z_{t,i}(\lambda) = \frac{S_t}{L_{t,i} - \lambda} = \frac{S_t}{L_{t-1,i} + \widetilde{\ell}_{t,i} - \lambda}. \tag{12}$$

Note that each $z_{t,i}(\lambda)$ is an increasing function of $\lambda$ on $(-\infty, L_{t,i}]$. Since $\widetilde{\ell}_{t,i_t} > 0$, it is easy to see that $z_{t,i_t}(Z_t) < x_{t,i_t}$. Thus in order to satisfy the sum-to-one constraint, we must have $\widetilde{Z}_t > Z_t$ and $\widetilde{Z}_t - Z_t < \widetilde{\ell}_{t,i_t}$. Moreover, we have $z_{t,i_t} < x_{t,i_t}$ and $z_{t,i} > x_{t,i}$ for all $i \ne i_t$. Therefore, we only need to prove that $z_{t,i_t} \ge \frac{1}{2}x_{t,i_t}$. Thanks to the monotonicity of $z_{t,i}(\lambda)$, it suffices to find a multiplier $\overline{Z}_t$ such that

$$z_{t,i_t}(\overline{Z}_t) \ge \frac{x_{t,i_t}}{2}, \tag{13}$$

$$\sum_{i \in [K]} z_{t,i}(\overline{Z}_t) \le 1. \tag{14}$$

With the above two conditions hold, we can conclude that $\widetilde{Z}_t \ge \overline{Z}_t$, *i.e.*, $\overline{Z}_t$ is a lower-bound for the actual multiplier $\widetilde{Z}_t$ making $z_t \in \Delta$, and thus $z_{t,i_t} = z_{t,i_t}(\widetilde{Z}_t) \ge z_{t,i_t}(\overline{Z}_t) \ge \frac{x_{t,i_t}}{2}$.

For our purpose, we will choose $\overline{Z}_t$ as follows

$$\overline{Z}_t = \begin{cases} Z_t & \text{if } x_{t,i_t} < \frac{1}{2}, \\ \text{the unique solution to } z_{t,i_t}(\lambda) = \frac{x_{t,i_t}}{2} & \text{if } x_{t,i_t} \ge \frac{1}{2}. \end{cases}$$

We will verify Eq. (13) and Eq. (14) for both cases. For the condition Eq. (13), when $x_{t,i_t} \ge \frac{1}{2}$, it automatically holds by definition of $\overline{Z}_t$. When $x_{t,i_t} < \frac{1}{2}$, we have $S_t = 4C_{t,i_t} \cdot (1 - x_{t,i_t}) \ge 2\ell_{t,i_t}^{\text{skip}}$. Therefore,

$$\frac{S_t}{L_{t,i_t} - \overline{Z}_t} = \frac{S_t}{L_{t-1,i_t} + \frac{\ell_{t,i_t}^{\text{skip}}}{x_{t,i_t}} - Z_t}$$

$$\geq \frac{S_t}{L_{t-1,i_t} + \frac{S_t}{2x_{t,i_t}} - Z_t}$$

$$= \frac{S_t}{(L_{t-1,i_t} - Z_t) \cdot (1 + \frac{S_t}{2x_{t,i_t}(L_{t-1,i_t} - Z_t)})}$$

$$= x_{t,i_t} \cdot \left(1 + \frac{1}{2x_{t,i_t}} \cdot x_{t,i_t}\right)^{-1}$$

$$= \frac{2}{3} x_{t,i_t}$$

$$\geq \frac{1}{2} x_{t,i_t}.$$

Thus, we have $\frac{S_t}{L_{t,i_t} - \overline{Z}_t} \geq \frac{x_{t,i_t}}{2}$ holds regardless $x_{t,i_t} \geq 1/2$ or not.

Then, we verify the other statement Eq. (14). When $x_{t,i_t} < \frac{1}{2}$, we have

$$\sum_{i \in [K]} z_{t,i}(\overline{Z}_t) = z_{t,i_t}(Z_t) + \sum_{i \neq i_t} z_{t,i}(Z_t)$$

$$= z_{t,i_t}(Z_t) + \sum_{i \neq i_t} x_{t,i}$$

$$< x_{t,i_t} + \sum_{i \neq i_t} x_{t,i} = 1.$$

For the other case $x_{t,i_t} \geq \frac{1}{2}$. It turns out that the definition of $\overline{Z}_t$, i.e., $z_{t,i_t}(\overline{Z}_t) = \frac{x_{t,i_t}}{2}$, solves to

$$\overline{Z}_t = -\frac{2S_t}{x_{t,i_t}} + L_{t-1,i_t} + \frac{\ell_{t,i_t}^{\text{skip}}}{x_{t,i_t}}$$

$$\leq Z_t - \frac{S_t}{x_{t,i_t}} + \frac{C_{t,i_t}}{x_{t,i_t}}$$

$$\leq Z_t - \frac{S_t}{x_{t,i_t}} + \frac{S_t}{4x_{t,i_t}^2(1 - x_{t,i_t})}$$

$$= Z_t + \frac{S_t(1 - 4x_{t,i_t}(1 - x_{t,i_t}))}{4x_{t,i_t}^2(1 - x_{t,i_t})}$$

$$= Z_t + \frac{S_t(2x_{t,i_t} - 1)^2}{4x_{t,i_t}^2(1 - x_{t,i_t})}$$

where the first inequality holds by the definition of $x_{t,i_t}$ in Eq. (12) and $0 < \ell_{t,i_t}^{\text{skip}} \leq C_{t,i_t} = \frac{S_t}{4(1-x_{t,i_t})} \leq \frac{S_t}{4x_{t,i_t}(1-x_{t,i_t})}$. For all $i \neq i_t$, we then have

$$\frac{S_t}{L_{t,i} - \overline{Z}_t} = \frac{S_t}{L_{t-1,i} - \overline{Z}_t}$$

$$\leq \frac{S_t}{L_{t-1,i} - Z_t - \frac{S_t(2x_{t,i_t}-1)^2}{4x_{t,i_t}^2(1-x_{t,i_t})}}$$

$$= \frac{S_t}{(L_{t-1,i} - Z_t) \cdot \left(1 - \frac{S_t(2x_{t,i_t}-1)^2}{(L_{t-1,i}-Z_t)4x_{t,i_t}^2(1-x_{t,i_t})}\right)}$$

$$= x_{t,i} \cdot \left(1 - x_{t,i} \frac{(2x_{t,i_t} - 1)^2}{4x_{t,i_t}^2(1 - x_{t,i_t})}\right)^{-1}$$

Since we have $x_{t,i} \leq \sum_{i \neq i_t} x_{t,i} = 1 - x_{t,i_t}$, then

$$z_{t,i}(\overline{Z}_t) = \frac{S_t}{L_{t,i} - \overline{Z}_t} \leq x_{t,i} \cdot \left(1 - \frac{1 - 4x_{t,i_t} + 4x_{t,i_t}^2}{4x_{t,i_t}^2}\right)^{-1} = x_{t,i} \cdot \frac{4x_{t,i_t}^2}{4x_{t,i_t} - 1}. \tag{15}$$

Therefore, we have

$$
\begin{aligned}
\sum_{i \in [K]} z_{t,i}(\overline{Z}_t) &\overset{(a)}{\leq} \frac{1}{2} x_{t,i_t} + \frac{4 x_{t,i_t}^2}{4 x_{t,i_t} - 1} \sum_{i \neq i_t} x_{t,i} \\
&= \frac{1}{2} x_{t,i_t} + \frac{4 x_{t,i_t}^2}{4 x_{t,i_t} - 1} (1 - x_{t,i_t}) \\
&\overset{(b)}{\leq} \frac{1}{2} x_{t,i_t} + 4 x_{t,i_t}^2 (1 - x_{t,i_t}) \\
&\overset{(c)}{\leq} 1,
\end{aligned}
$$

where step $(a)$ is obtained by applying Eq. (15) to $i \neq i_t$ and $z_{t,i_t}(\overline{Z}_t) = x_{t,i_t}/2$; step $(b)$ is due to $x_{t,i_t} \geq 1/2$; step $(c)$ is due to the fact that $x \mapsto x/2 + 4x^2(1 - x)$ has a maximum value less than 1. Then, we have already verified Eq. (14), which finishes the proof of $z_{t,i_t} \geq x_{t,i_t}/2$.

**The right side $z_{t,i} \leq 2 x_{t,i}$.** We then show that $z_{t,i} \leq 2 x_{t,i}$ holds for all $i \in [K]$. The main idea is similar to what we have done in the argument for the left-side inequality, we will find some $\overline{Z}_t$ under which we can verify that

$$
z_{t,i}(\overline{Z}_t) \leq 2 x_{t,i} \quad \forall i \neq i_t, \tag{16}
$$

$$
\sum_{i \in [K]} z_{t,i}(\overline{Z}_t) \geq 1. \tag{17}
$$

We can then claim that this $\overline{Z}_t$ is indeed an upper-bound of the actual multiplier $\widetilde{Z}_t$.

Let $j^* = \operatorname{argmax}_{j \neq i_t} x_{t,j}$, we just take $\overline{Z}_t$ to be the unique solution to $z_{t,j^*}(\lambda) = 2 x_{t,j^*}$, which solves to

$$
\overline{Z}_t = Z_t + \frac{S_t}{2 x_{t,j^*}}.
$$

One can verify that

$$
\begin{aligned}
z_{t,j^*}(\overline{Z}_t) &= \frac{S_t}{(L_{t-1,j^*} - Z_t) \cdot \left(1 - \frac{S_t}{(L_{t-1,j^*} - Z_t) \cdot 2 x_{t,j^*}}\right)} \\
&= x_{t,j^*} \left(1 - \frac{x_{t,j^*}}{2 x_{t,j^*}}\right)^{-1} \\
&= 2 x_{t,j^*}.
\end{aligned}
$$

For $i \in [K] \setminus \{i_t, j^*\}$, it is easy to see that

$$
\begin{aligned}
z_{t,i}(\overline{Z}_t) &= \frac{S_t}{L_{t-1,i} - Z_t - \frac{S_t}{2 x_{t,j^*}}} \\
&= \frac{S_t}{(L_{t-1,i} - Z_t) \cdot \left(1 - \frac{S_t}{2 x_{t,j^*}(L_{t-1,i} - Z_t)}\right)} \\
&= x_{t,i} \left(1 - \frac{x_{t,i}}{2 x_{t,j^*}}\right)^{-1} \\
&\leq 2 x_{t,i}.
\end{aligned}
$$

Hence Eq. (16) holds. In order to verify Eq. (17), note that

$$
\widetilde{\ell}_{t,i_t} \leq \frac{C_{t,i_t}}{x_{t,i_t}} \leq \frac{S_t}{4 x_{t,i_t}(1 - x_{t,i_t})}.
$$

When $x_{t,i_t} \geq 1/2$, we have

$$
\widetilde{\ell}_{t,i_t} \leq \frac{S_t}{4 \cdot \frac{1}{2} \cdot \sum_{j \neq i_t} x_{t,j}}
$$

$$\leq \frac{S_t}{2x_{t,j^*}}$$
$$= \overline{Z}_t - Z_t.$$

Therefore, we have

$$z_{t,i_t}(\overline{Z}_t) = \frac{S_t}{L_{t,i_t} - \overline{Z}_t} \geq \frac{S_t}{L_{t-1,i_t} + \overline{Z}_t - Z_t - \overline{Z}_t} = x_{t,i_t}.$$

Therefore,

$$\sum_{i \in [K]} z_{t,i}(\overline{Z}_t) = z_{t,i_t}(\overline{Z}_t) + \sum_{i \neq i_t} z_{t,i}(\overline{Z}_t) \tag{18}$$

$$\geq z_{t,i_t}(\overline{Z}_t) + \sum_{i \neq i_t} z_{t,i}(Z_t)$$

$$\geq x_{t,i_t} + \sum_{i \neq i_t} x_{t,i} = 1.$$

On the other hand, when $x_{t,i_t} < 1/2$, we have

$$\widetilde{\ell}_{t,i_t} \leq \frac{S_t}{4x_{t,i_t}(1 - x_{t,i_t})} \leq \frac{S_t}{4x_{t,i_t} \cdot \frac{1}{2}} = \frac{S_t}{2x_{t,i_t}}.$$

If $x_{t,i_t} \geq x_{t,j^*}$, we have $\widetilde{\ell}_{t,i_t} \leq S_t/(2x_{t,j^*}) = \overline{Z}_t - Z_t$. Then we can apply the same analysis as Eq. (18). Hence the only remaining case is $x_{t,i_t} < 1/2$ and $x_{t,i_t} < x_{t,j^*}$, where we have

$$z_{t,j^*}(\overline{Z}_t) - x_{t,j^*} = x_{t,j^*} > x_{t,i_t} > x_{t,i_t} - z_{t,i_t}(\overline{Z}_t). \tag{19}$$

Therefore, we have

$$\sum_{i \in [K]} z_{t,i}(\overline{Z}_t) - 1 = \left[ z_{t,j^*}(\overline{Z}_t) - x_{t,j^*} \right] + \left[ z_{t,i_t}(\overline{Z}_t) - x_{t,i_t} \right] + \sum_{i \neq i_t, j^*} z_{t,i}(\overline{Z}_t) - x_{t,i}$$

$$> \sum_{i \neq i_t, j^*} z_{t,i}(\overline{Z}_t) - x_{t,i}$$

$$\geq 0$$

hence $\sum_{i \in [K]} z_{t,i}(\overline{Z}_t) > 1$. In all, we show that $\sum_{i \in [K]} z_{t,i}(\overline{Z}_t) \geq 1$, and we are done.

**Negative Loss.** The proof of case $\widetilde{\ell}_{t,i} < 0$ is very similar. In this case, we have $z_{t,i_t}(Z_t) > x_{t,i_t}$, which shows that $\sum_{i \in [K]} z_{t,i}(Z_t) > 1$. Therefore, we have $\widetilde{Z}_t < Z_t$, which implies $z_{t,i} < x_{t,i}$ for $i \neq i_t$ and $z_{t,i_t} > x_{t,i_t}$. Therefore, we only need to verify that $z_{t,i_t} \leq 2x_{t,i_t}$ and $z_{t,i} \geq \frac{1}{2}x_{t,i}$ for $i \neq i_t$. We apply similar proof process as above statement.

For the first inequality, we just need to verify that there exists a multiplier $\overline{Z}_t$ such that

$$z_{t,i_t}(\overline{Z}_t) \leq 2x_{t,i_t}, \tag{20}$$

$$\sum_{i \in [K]} z_{t,i}(\overline{Z}_t) \geq 1. \tag{21}$$

We set $\overline{Z}_t$ as the unique solution to $z_{t,i_t}(\lambda) = 2x_{t,i_t}$. Then we have

$$\overline{Z}_t = -\frac{S_t}{2x_{t,i_t}} + L_{t-1,i_t} + \widetilde{\ell}_{t,i_t}$$

$$= -\frac{S_t}{x_{t,i_t}} + \frac{S_t}{2x_{t,i_t}} + L_{t-1,i_t} + \frac{\ell_{t,i_t}^{\text{skip}}}{x_{t,i_t}}$$

$$\geq Z_t + \frac{S_t}{2x_{t,i_t}} - \frac{C_{t,i_t}}{x_{t,i_t}}$$

$$= Z_t + \frac{S_t}{2x_{t,i_t}} - \frac{S_t}{4x_{t,i_t}(1 - x_{t,i_t})}$$

$$= Z_t - \frac{S_t(2x_{t,i_t} - 1)}{4x_{t,i_t}(1 - x_{t,i_t})}$$

If $x_{t,i_t} \le 1/2$, we have $\overline{Z}_t \ge Z_t$. Therefore,

$$\sum_{i \in [K]} z_{t,i}(\overline{Z}_t) \ge z_{t,i_t}(\overline{Z}_t) + \sum_{i \ne i_t} z_{t,i}(Z_t) = 2x_{t,i_t} + \sum_{i \ne i_t} x_{t,i} \ge 1.$$

For the other case, if $x_{t,i_t} > 1/2$, we have for any $i \ne i_t$

$$z_{t,i}(\overline{Z}_t) \ge \frac{S_t}{L_{t-1,i} - Z_t + \frac{S_t(2x_{t,i_t}-1)}{4x_{t,i_t}(1-x_{t,i_t})}}$$

$$= x_{t,i} \cdot \left(1 + x_{t,i} \cdot \frac{2x_{t,i_t} - 1}{4x_{t,i_t}(1 - x_{t,i_t})}\right)^{-1}$$

$$\ge x_{t,i} \cdot \left(1 + \frac{2x_{t,i_t} - 1}{4x_{t,i_t}}\right)^{-1}$$

$$= x_{t,i} \cdot \frac{4x_{t,i_t}}{6x_{t,i_t} - 1},$$

where the second inequality holds by $2x_{t,i_t} - 1 > 0$ and $x_{t,i} \le 1 - x_{t,i_t}$. Therefore, we have

$$\sum_{i \in [K]} z_{t,i}(\overline{Z}_t) \ge 2x_{t,i_t} + \sum_{i \ne i_t} x_{t,i} \cdot \frac{4x_{t,i_t}}{6x_{t,i_t} - 1} = 2x_{t,i_t} + \frac{4x_{t,i_t}(1 - x_{t,i_t})}{6x_{t,i_t} - 1} \ge 1.$$

For the second inequality, we need to verify that there exists a multiplier $\overline{Z}_t$ such that

$$z_{t,i}(\overline{Z}_t) \ge \frac{1}{2}x_{t,i} \quad \forall i \ne i_t, \tag{22}$$

$$\sum_{i \in [K]} z_{t,i}(\overline{Z}_t) \le 1. \tag{23}$$

Let $j^* = \operatorname{argmax}_{j \ne i_t} x_{t,j}$. Then we set $\overline{Z}_t$ as

$$\overline{Z}_t = Z_t - \frac{S_t}{x_{t,j^*}}.$$

Hence, we have for any $i \ne i_t$,

$$z_{t,i}(\overline{Z}_t) = \frac{S_t}{(L_{t-1,i} - Z_t) \cdot \left(1 + \frac{S_t}{(L_{t-1,i} - Z_t)x_{t,j^*}}\right)}$$

$$= x_{t,i} \cdot \left(1 + \frac{x_{t,i}}{x_{t,j^*}}\right)^{-1}$$

$$\ge x_{t,i} \cdot \frac{1}{2},$$

where the inequality holds by $x_{t,j^*} > x_{t,i}$. Therefore, we have

$$z_{t,i}(\overline{Z}_t) \ge \frac{1}{2}x_{t,i}.$$

Then we verity Eq. (23). Since we have $\overline{Z}_t \le Z_t$, then $z_{t,i}(\overline{Z}_t) \le x_{t,i}$ for any $i \ne i_t$. Thus we only need to prove $z_{t,i_t}(\overline{Z}_t) \le x_{t,i_t}$. Notice that

$$z_{t,i_t}(\overline{Z}_t) = \frac{S_t}{L_{t-1,i_t} - Z_t + \widetilde{\ell}_{t,i_t} + \frac{S_t}{x_{t,j^*}}} \le \frac{S_t}{L_{t-1,i_t} - Z_t + \frac{S_t}{x_{t,j^*}} - \frac{S_t}{4x_{t,i_t}(1-x_{t,i_t})}},$$

where the inequality is due to $\widetilde{\ell}_{t,i_t} = \ell_{t,i_t}^{\mathrm{skip}}/x_{t,i_t} \ge -C_{t,i_t}/x_{t,i_t}$ and the definition of $C_{t,i_t}$ in Eq. (3). If we have $x_{t,i_t} \ge 1/2$, then

$$\frac{S_t}{x_{t,j^*}} - \frac{S_t}{4x_{t,i_t}(1 - x_{t,i_t})} \ge \frac{S_t}{x_{t,j^*}} - \frac{S_t}{2(1 - x_{t,i_t})}$$

$$= \frac{S_t}{x_{t,j^*}} - \frac{S_t}{2\sum_{j\neq i_t} x_{t,j}}$$

$$\geq \frac{S_t}{x_{t,j^*}} - \frac{S_t}{2x_{t,j^*}}$$

$$\geq 0.$$

Therefore, we have $z_{t,i_t}(\overline{Z}_t) \leq x_{t,i_t}$, which implies that $\sum_{i\in[K]} z_{t,i}(\overline{Z}_t) \leq \sum_{i\in[K]} x_{t,i} = 1$. For the case when $x_{t,i_t} < 1/2$, we also have

$$\frac{S_t}{x_{t,j^*}} - \frac{S_t}{4x_{t,i_t}(1 - x_{t,i_t})} \geq \frac{S_t}{x_{t,j^*}} - \frac{S_t}{2x_{t,i_t}}.$$

If $x_{t,j^*} \leq 2x_{t,i_t}$, we still have $z_{t,i_t}(\overline{Z}_t) \leq x_{t,i_t}$. Otherwise, for $x_{t,j^*} > 2x_{t,i_t}$, we can write

$$z_{t,i_t}(\overline{Z}_t) = \frac{S_t}{L_{t-1,i_t} - Z_t + \widetilde{\ell}_{t,i_t} + \frac{S_t}{x_{t,j^*}}}$$

$$\leq \frac{S_t}{L_{t-1,i_t} - Z_t + \frac{S_t}{x_{t,j^*}} - \frac{S_t}{4x_{t,i_t}(1-x_{t,i_t})}}$$

$$= x_{t,i_t} \cdot \left(1 + \frac{x_{t,i_t}}{x_{t,j^*}} - \frac{1}{4(1 - x_{t,i_t})}\right)^{-1}$$

Notice that here $x_{t,i_t} \in [0, 1/2]$, which implies that $1 + \frac{x_{t,i_t}}{x_{t,j^*}} - \frac{1}{4(1-x_{t,i_t})} \geq 1/2$. Therefore, we have

$$z_{t,i_t}(\overline{Z}_t) \leq 2x_{t,i_t} \leq \frac{1}{2}x_{t,j^*} + x_{t,i_t}.$$

Since $z_{t,j^*}(\overline{Z}_t) = \frac{1}{2}x_{t,j^*}$, we have

$$\sum_{i\in[K]} z_{t,i}(\overline{Z}_t) \leq \sum_{i\in[K]\setminus\{i_t,j^*\}} x_{t,i} + \frac{1}{2}x_{t,j^*} + \frac{1}{2}x_{t,j^*} + x_{t,i_t} = 1.$$

In all, we conclude Eq. (23) and prove this lemma. $\square$

## C.2 CALCULATING $w_{t,i}$ VIA PARTIAL DERIVATIVES

By inspecting the partial derivatives of the multiplier $\widetilde{Z}_t$ with respect to the feedback vector $\widetilde{\ell}_t$, we can derive the following lemmas on $w_{t,i}$.

**Lemma 16.** *We have*

$$w_{t,i} = \widetilde{\ell}_{t,i_t} \cdot \frac{x_{t,i}}{S_t} \cdot \left(\mathbb{1}[i = i_t] - \frac{\zeta_{t,i_t}^2}{\sum_{k\in[K]} \zeta_{t,k}^2}\right),$$

*where*

$$\zeta_t := \underset{\boldsymbol{z}\in\Delta}{\operatorname{argmin}}\langle \boldsymbol{L}_{t-1} + \lambda\widetilde{\ell}_t, \boldsymbol{z}\rangle + \Psi_t(\boldsymbol{z}).$$

*Here $\lambda$ is some constant ranged from $(0, 1)$, implicitly determined by $\widetilde{\ell}_t$.*

*Proof.* Notice that conditioning on the time step before $t - 1$, $z_{t,i}$ only depends on $\widetilde{\ell}_{t,i}$. Then we denote

$$\widetilde{Z}'_{t,i}(\theta) := \left.\frac{\partial \widetilde{Z}_t}{\partial \widetilde{\ell}_{t,i}}\right|_{\widetilde{\ell}_t = \theta\boldsymbol{e}_i},$$

where $\boldsymbol{e}_i$ is the unit vector supported at the $i$-th coordinate, and $\theta$ is a scalar.

Notice that

$$\sum_{j \in [K]} z_{t,j} = \sum_{j \in [K]} \frac{S_t}{L_{t,j} - \widetilde{Z}_t} = 1, \tag{24}$$

Then partially derivate $\widetilde{\ell}_{t,i}$ on both sides of Eq. (24), we get

$$\sum_{j \in [K]} -\frac{S_t}{(\ell_{t,j} - \widetilde{Z}_t)^2} \cdot (\mathbb{1}[j = i] - \widetilde{Z}'_{t,i}) = -\frac{z_{t,i}^2}{S_t} + \widetilde{Z}_{t,i}^{\text{skip}} \sum_{j \in [K]} \frac{z_{t,j}^2}{S_t} = 0,$$

which solves to

$$\widetilde{Z}'_{t,i} = \frac{z_{t,i}^2}{\sum_{j \in [K]} z_{t,j}^2}.$$

Similarly, we denote

$$z'_{t,ij}(\theta) := \left. \frac{\partial z_{t,i}}{\partial \widetilde{\ell}_{t,j}} \right|_{\widetilde{\boldsymbol{\ell}}_t = \theta \boldsymbol{e}_j}$$

Then according to the chain rule, we have

$$z'_{t,ij} = \frac{z_{t,i}^2}{S_t} \cdot \left( z_{t,j}^2 / \left( \sum_{k \in [K]} x_{t,k}^2 \right) - \mathbb{1}[i = j] \right).$$

We denote

$$w'_{t,ij}(\theta) := \left. \frac{\partial w_{t,i}}{\partial \widetilde{\ell}_{t,j}} \right|_{\widetilde{\boldsymbol{\ell}}_t = \theta \boldsymbol{e}_j}.$$

Recall that $w_{t,i} = \frac{x_{t,i}}{S_t}(\widetilde{\ell}_{t,i} - (\widetilde{Z}_t - Z_t))$, hence

$$w'_{t,ij} = \frac{x_{t,i}}{S_t} \cdot \left( \mathbb{1}[i = j] - \left( \frac{z_{t,j}^2}{S_t} \right) / \left( \sum_{k \in [K]} \frac{z_{t,k}^2}{S_t} \right) \right)$$

$$= \frac{x_{t,i}}{S_t} \cdot \left( \mathbb{1}[i = j] - (z_{t,j}^2) / \left( \sum_{k \in [K]} z_{t,k}^2 \right) \right).$$

Thus according to the intermediate value theorem, we have

$$w_{t,i} = \widetilde{\ell}_{t,i_t} \cdot \frac{x_{t,i}}{S_t} \cdot \left( \mathbb{1}[i = i_t] - \frac{\zeta_{t,i_t}^2}{\sum_{k \in [K]} \zeta_{t,k}^2} \right),$$

where

$$\zeta_t := \operatorname*{argmin}_{\boldsymbol{z} \in \Delta^{[K]}} \langle \boldsymbol{L}_{t-1} + \lambda \widetilde{\ell}_t, \boldsymbol{z} \rangle + \Psi_t(\boldsymbol{z}),$$

and $\lambda$ is some constant ranged from $(0, 1)$, implicitly determined by $\widetilde{\ell}_t$. Furthermore, Lemma 4 guarantees that

$$x_{t,i}/2 \le \zeta_{t,i} \le 2x_{t,i} \quad \forall i \in [K]. \tag{25}$$

$\square$

### C.3 Bregman Divergence before Expectation: Proof of Lemma 5

**Lemma 17** (Formal version of Lemma 5). *We have*

$$\text{DIV}_t \leq 2048 \cdot S_t^{-1}(\ell_{t,i_t}^{skip})^2(1 - x_{t,i_t})^2.$$

*Therefore, the expectation of the Bregman Divergence term $\text{DIV}_t$ can be bounded by*

$$\mathbb{E}[\text{DIV}_t \mid \mathcal{F}_{t-1}] \leq 2048 \sum_{i \in [K]} S_t^{-1} \mathbb{E}[(\ell_{t,i}^{skip})^2 \mid \mathcal{F}_{t-1}] x_{t,i}(1 - x_{t,i})^2.$$

*Moreover, for any $\mathcal{T} \in [T]$, we can bound the sum of Bregman divergence term by*

$$\sum_{t=1}^{\mathcal{T}} \text{DIV}_t \leq 4096 \cdot S_{\mathcal{T}+1} K \log T.$$

*Proof.* Recall the definition of $w_{t,i}$, we have

$$\text{DIV}_t = \sum_{i \in [K]} S_t(w_{t,i} - \log(1 + w_{t,i}))$$

By Lemma 4, we have $w_{t,i} = x_{t,i}/z_{t,i} - 1 \in [-1/2, 1/2]$. Therefore, since $x - \log(1 + x) \leq x^2$, we have

$$\text{DIV}_t \leq \sum_{i \in [K]} S_t \cdot \left( \mathbb{1}[i = i_t] w_{t,i}^2 + \mathbb{1}[i \neq i_t] w_{t,i}^2 \right). \tag{26}$$

By Lemma 16, we have

$$w_{t,i} = \widetilde{\ell}_{t,i_t} \cdot \frac{x_{t,i}}{S_t} \cdot \left( \mathbb{1}[i = i_t] - \frac{\zeta_{t,i_t}^2}{\sum_{k \in [K]} \zeta_{t,k}^2} \right),$$

and $\zeta_{t,i}$ satisfying $x_{t,i}/2 \leq \zeta_{t,i} \leq 2x_{t,i}$ by Eq. (25). Then, for $i \neq i_t$, we have

$$w_{t,i} = -\ell_{t,i_t}^{skip} \cdot \frac{x_{t,i}}{S_t} \cdot \frac{\zeta_{t,i_t}}{\sum_{k \in [K]} \zeta_{t,k}^2},$$

$$w_{t,i}^2 = (\ell_{t,i_t}^{skip})^2 \cdot \frac{x_{t,i}^2}{S_t^2} \cdot \frac{\zeta_{t,i_t}^2}{\left( \sum_{k \in [K]} \zeta_{t,k}^2 \right)^2} \leq 16 \cdot (\ell_{t,i_t}^{skip})^2 \cdot \frac{x_{t,i}^2}{S_t^2} \cdot \frac{x_{t,i_t}^2}{\left( \sum_{k \in [K]} x_{t,k}^2 \right)^2}.$$

And for $i = i_t$,

$$w_{t,i_t} = \ell_{t,i_t}^{skip} \cdot \frac{1}{S_t} \cdot \frac{\sum_{j \neq i_t} \zeta_{t,j}^2}{\sum_{k \in [K]} \zeta_{t,k}^2},$$

$$w_{t,i_t}^2 = (\ell_{t,i_t}^{skip})^2 \cdot \frac{1}{S_t^2} \cdot \frac{\left( \sum_{j \neq i_t} \zeta_{t,j}^2 \right)^2}{\left( \sum_{k \in [K]} \zeta_{t,k}^2 \right)^2} \leq 256 \cdot (\ell_{t,i_t}^{skip})^2 \cdot \frac{1}{S_t^2} \cdot \frac{\left( \sum_{j \neq i_t} x_{t,j}^2 \right)^2}{\left( \sum_{k \in [K]} x_{t,k}^2 \right)^2}.$$

Therefore, we have

$$\sum_{i \in [K]} S_t \cdot \left( \mathbb{1}[i = i_t] w_{t,i}^2 + \mathbb{1}[i \neq i_t] w_{t,i}^2 \right)$$

$$\leq 256 \cdot (\ell_{t,i_t}^{skip})^2 \cdot S_t^{-1} \cdot \left( \frac{\left( \sum_{j \neq i_t} x_{t,j}^2 \right)^2}{\left( \sum_{k \in [K]} x_{t,k}^2 \right)^2} + \sum_{i \neq i_t} \frac{x_{t,i}^2 \cdot x_{t,i_t}^2}{\left( \sum_{k \in [K]} x_{t,k}^2 \right)^2} \right) \tag{27}$$

Notice that

$$\frac{\left( \sum_{j \neq i_t} x_{t,j}^2 \right)^2}{\left( \sum_{k \in [K]} x_{t,k}^2 \right)^2} + \sum_{i \neq i_t} \frac{x_{t,i}^2 \cdot x_{t,i_t}^2}{\left( \sum_{k \in [K]} x_{t,k}^2 \right)^2} = \left( 1 - \frac{x_{t,i_t}^2}{\sum_{k \in [K]} x_{t,k}^2} \right)^2 + \frac{\sum_{i \neq i_t} x_{t,i}^2}{\sum_{k \in [K]} x_{t,k}^2} \cdot \frac{x_{t,i_t}^2}{\sum_{k \in [K]} x_{t,k}^2}.$$

For $x_{t,i_t} \leq 1/2$, we have $4(1 - x_{t,i_t})^2 \geq 1$. Then we have

$$\left(1 - \frac{x_{t,i_t}^2}{\sum_{k\in[K]} x_{t,k}^2}\right)^2 + \frac{\sum_{i\neq i_t} x_{t,i}^2}{\sum_{k\in[K]} x_{t,k}^2} \cdot \frac{x_{t,i_t}^2}{\sum_{k\in[K]} x_{t,k}^2} \leq 1 + 1 \cdot 1 \leq 8(1 - x_{t,i_t})^2. \quad (28)$$

For $x_{t,i_t} \geq 1/2$, we denote

$$\widetilde{x}_{t,i} := \frac{x_{t,i}^2}{\sum_{k\in[K]} x_{t,k}^2}, \quad \forall i \in [K],$$

which satisfies $\widetilde{x}_{t,i} \leq x_{t,i}^2/x_{t,i_t}^2 \leq 4x_{t,i}^2$ for every $i \in [K]$. Since $x_{t,i_t} \geq 1/2$, we also have $1/2 \leq x_{t,i_t} \leq \widetilde{x}_{t,i_t} \leq 1$. Therefore,

$$\begin{aligned}
\left(1 - \frac{x_{t,i_t}^2}{\sum_{k\in[K]} x_{t,k}^2}\right)^2 + \frac{\sum_{i\neq i_t} x_{t,i}^2}{\sum_{k\in[K]} x_{t,k}^2} \cdot \frac{x_{t,i_t}^2}{\sum_{k\in[K]} x_{t,k}^2} &= (1 - \widetilde{x}_{t,i_t})^2 + \widetilde{x}_{t,i_t} \sum_{i\neq i_t} \widetilde{x}_{t,i} \\
&\leq (1 - x_{t,i_t})^2 + \sum_{i\neq i_t} 4x_{t,i}^2 \\
&\leq (1 - x_{t,i_t})^2 + 4\left(\sum_{i\neq i_t} x_{t,i}\right)^2 \\
&\leq 5(1 - x_{t,i_t})^2
\end{aligned} \quad (29)$$

Combine these cases, we get

$$\begin{aligned}
\text{DIV}_t &\leq \sum_{i\in[K]} S_t \cdot \left(\mathbb{1}[i = i_t] w_{t,i}^2 + \mathbb{1}[i \neq i_t] w_{t,i}^2\right) \\
&\leq 256 \cdot (\ell_{t,i_t}^{\text{skip}})^2 \cdot S_t^{-1} \cdot \left(\left(1 - \frac{x_{t,i_t}^2}{\sum_{k\in[K]} x_{t,k}^2}\right)^2 + \frac{\sum_{i\neq i_t} x_{t,i}^2}{\sum_{k\in[K]} x_{t,k}^2} \cdot \frac{x_{t,i_t}^2}{\sum_{k\in[K]} x_{t,k}^2}\right) \\
&\leq 2048 \cdot (\ell_{t,i_t}^{\text{skip}})^2 \cdot S_t^{-1} \cdot (1 - x_{t,i_t})^2,
\end{aligned}$$

where the first inequality is due to Eq. (26), the second inequality is due to Eq. (27), and the last inequality is due to Eq. (28) for $x_{t,i_t} \leq 1/2$ and Eq. (29) for $x_{t,i_t} \geq 1/2$.

Moreover, taking expectation conditioning on $\mathcal{F}_{t-1}$, we directly imply

$$\mathbb{E}[\text{DIV}_t \mid \mathcal{F}_{t-1}] \leq 2048 \sum_{i\in[K]} S_t^{-1} \mathbb{E}[(\ell_{t,i}^{\text{skip}})^2 \mid \mathcal{F}_{t-1}] x_{t,i}(1 - x_{t,i})^2.$$

Consider the sum of $\text{DIV}_t$, we can write

$$\sum_{t=1}^{\mathcal{T}} \text{DIV}_t \leq 2048 \sum_{t=1}^{\mathcal{T}} S_t^{-1} \cdot (\ell_{t,i_t}^{\text{skip}})^2 \cdot (1 - x_{t,i_t})^2.$$

Notice that by definition of $S_{t+1}$ in Eq. (4), we have

$$(K \log T) \cdot (S_{t+1}^2 - S_t^2) = \left(\ell_{t,i_t}^{\text{clip}}\right)^2 \cdot (1 - x_{t,i_t})^2.$$

Moreover, since $|\ell_{t,i_t}^{\text{clip}}|$ is controlled by $C_{t,i_t}$, we have

$$S_{t+1}^2 \leq S_t^2 + C_{t,i_t}^2 (1 - x_{t,i_t})^2 \cdot (K \log T)^{-1} = S_t^2 \left(1 + (4K \log T)^{-1}\right) \leq 2S_t^2, \quad (30)$$

where the first inequlity is due to $|\ell_{t,i_t}^{\text{clip}}| \leq C_{t,i_t}$ and the definition of $C_{t,i_t}$ in Eq. (3), and the second inequality holds by $T \geq 2$. Therefore, as $|\ell_{t,i_t}^{\text{skip}}| \leq |\ell_{t,i_t}^{\text{clip}}|$ trivially, we have

$$\sum_{t=1}^{\mathcal{T}} \text{DIV}_t \leq 2048 \sum_{t=1}^{T} S_t^{-1} \cdot (\ell_{t,i_t}^{\text{clip}})^2 (1 - x_{t,i_t})^2$$

$$\leq 2048 \cdot K \log T \sum_{t=1}^{\mathcal{T}} \frac{S_{t+1}^2 - S_t^2}{S_t}$$

$$= 2048 \cdot K \log T \sum_{t=1}^{\mathcal{T}} \frac{(S_{t+1} + S_t)(S_{t+1} - S_t)}{S_t}$$

$$\leq 2048(1 + \sqrt{2}) K \log T \sum_{t=1}^{\mathcal{T}} S_{t+1} - S_t$$

$$= 4096 \cdot S_{\mathcal{T}+1} K \log T,$$

where the first inequality is by Lemma 5 and $|\ell_{t,i_t}^{\text{skip}}| \leq |\ell_{t,i_t}^{\text{clip}}|$, the second inequality is due to the definition of $S_{t+1}$ in Eq. (4) and $|\ell_{t,i_t}^{\text{skip}}| \leq |\ell_{t,i_t}^{\text{clip}}|$, and the last inequality is due to the $S_{t+1} \leq \sqrt{2}S_t$ by Eq. (30). □

### C.4 BOUNDING THE EXPECTATION OF LEARNING RATE $S_{T+1}$: PROOF OF LEMMA 6

Recall the definition of $S_t$ in Eq. (31), we have

$$S_{T+1} = \sqrt{4 + \sum_{t=1}^{T} (\ell_{t,i_t}^{\text{clip}})^2 \cdot (1 - x_{t,i_t})^2 \cdot (K \log T)^{-1}}$$

$$\leq \sqrt{4 + \sum_{t=1}^{T} (\ell_{t,i_t}^{\text{clip}})^2 \cdot (K \log T)^{-1}}.$$

Therefore, to control $\mathbb{E}[S_{T+1}]$, it suffices to control $\mathbb{E}[S]$ where $S$ is defined as follows:

$$S := \sqrt{4 + \sum_{t=1}^{T} (\ell_{t,i_t}^{\text{clip}})^2 \cdot (K \log T)^{-1}}. \tag{31}$$

The formal proof of Lemma 6 which bounds $\mathbb{E}[S_{T+1}]$ is given below.

**Lemma 18** (Restatement of Lemma 6). *We have*

$$\mathbb{E}[S_{T+1}] \leq 2 \cdot \sigma (K \log T)^{-\frac{1}{\alpha}} T^{\frac{1}{\alpha}}. \tag{32}$$

*Proof.* By clipping operation, we have

$$|\ell_{t,i_t}^{\text{clip}}| \leq C_{t,i_t} = S_t \cdot \frac{1}{4(1 - x_{t,i_t})} \leq S_t \leq S_{T+1} \leq S.$$

Therefore, we know

$$4 + \sum_{t=1}^{T} (\ell_{t,i_t}^{\text{clip}})^2 \cdot (K \log T)^{-1} \leq \frac{1}{4} \sum_{t=1}^{T} |\ell_{t,i_t}|^\alpha \cdot S^{2-\alpha} \cdot (K \log T)^{-1},$$

where the inequality is given by $S^2 \geq 16$, $|\ell_{t,i_t}^{\text{clip}}| \leq S$ and $|\ell_{t,i_t}^{\text{clip}}| \leq |\ell_{t,i_t}|$. Hence,

$$S^\alpha \leq \frac{4}{3} (K \log T)^{-1} \sum_{t=1}^{T} |\ell_{t,i_t}|^\alpha.$$

Take expectation on both sides and use the convexity of mapping $x \mapsto x^\alpha$. We get

$$\mathbb{E}[S] \leq \left(\frac{4}{3}\right)^{1/\alpha} (K \log T)^{-1/\alpha} \cdot \sigma \cdot T^{1/\alpha}$$

$$\leq 2(K \log T)^{-1/\alpha} \cdot \sigma \cdot T^{1/\alpha}.$$

The conclusion follows from the fact that $S_{T+1} \leq S$. □

## C.5 ADVERSARIAL BOUNDS FOR BREGMAN DIVERGENCE TERMS: PROOF OF THEOREM 7

Equipped with Lemma 6, we can verify the main result (Theorem 7) for Bregman divergence terms in adversarial case.

**Theorem 19** (Formal version of Theorem 7). *We have*

$$\mathbb{E}\left[\sum_{t=1}^{T} \text{DIV}_t\right] \leq 8192 \cdot \sigma K^{1-1/\alpha} T^{1/\alpha} (\log T)^{1-1/\alpha}$$

*Proof.* By Lemma 17 and Lemma 6, we have

$$\mathbb{E}\left[\sum_{t=1}^{T} \text{DIV}_t\right] \leq 4096 \cdot \mathbb{E}[S_{T+1}] \cdot K \log T$$

$$\leq 8192 \cdot \sigma K^{1-1/\alpha} T^{1/\alpha} (\log T)^{1-1/\alpha}.$$

Ignoring the logarithmic terms, we will get $\mathbb{E}\left[\sum_{t=1}^{T} \text{DIV}_t\right] \leq \widetilde{\mathcal{O}}(\sigma K^{1-1/\alpha} T^{1/\alpha})$. $\qquad\square$

## C.6 STOCHASTIC BOUNDS FOR BREGMAN DIVERGENCE TERMS: PROOF OF THEOREM 8

We first calculate the expectation of a single Bregman divergence term $\mathbb{E}[\text{DIV}_t \mid \mathcal{F}_{t-1}]$.

**Lemma 20.** *Conditioning on $\mathcal{F}_{t-1}$, the expectation of Bregman divergence term $\text{DIV}_t$ can be bounded by*

$$\mathbb{E}[\text{DIV}_t \mid \mathcal{F}_{t-1}] \leq 4096 \cdot S_t^{1-\alpha} \sigma^{\alpha} (1 - x_{t,i^*}).$$

*Proof.* From Lemma 17, we have

$$\mathbb{E}[\text{DIV}_t \mid \mathcal{F}_{t-1}] \leq 2048 \cdot S_t^{-1} \mathbb{E}\left[((\ell_{t,i_t}^{\text{skip}})^2 (1 - x_{t,i_t})^2 \mid \mathcal{F}_{t-1}\right] \tag{33}$$

$$\leq 2048 \cdot S_t^{-1} \mathbb{E}\left[(\ell_{t,i_t}^{\text{clip}})^2 (1 - x_{t,i_t})^2 \mid \mathcal{F}_{t-1}\right] \tag{34}$$

By definition of $C_{t,i_t}$ in Eq. (3), we have

$$\mathbb{E}[\text{DIV}_t \mid \mathcal{F}_{t-1}] \leq 2048 \cdot S_t^{-1} \mathbb{E}\left[C_{t,i_t}^{2-\alpha} (1 - x_{t,i_t})^{2-\alpha} |\ell_{t,i_t}|^{\alpha} (1 - x_{t,i_t})^{\alpha}\right]$$

$$\leq 2048 \cdot \frac{S_t^{1-\alpha}}{4^{2-\alpha}} \sigma^{\alpha} \mathbb{E}[(1 - x_{t,i_t})^{\alpha} \mid \mathcal{F}_{t-1}]$$

$$\leq 2048 \cdot S_t^{1-\alpha} \sigma^{\alpha} \sum_{i \in [K]} x_{t,i} (1 - x_{t,i})^{\alpha}$$

Notice that

$$\sum_{i \in [K]} x_{t,i} (1 - x_{t,i})^{\alpha} \leq \sum_{i \neq i^*} x_{t,i} + x_{t,i^*} (1 - x_{t,i^*})$$

$$= (1 - x_{t,i^*}) + x_{t,i^*} (1 - x_{t,i^*})$$

$$\leq 2(1 - x_{t,i^*})$$

Therefore, we have

$$\mathbb{E}[\text{DIV}_t \mid \mathcal{F}_{t-1}] \leq 4096 \cdot S_t^{1-\alpha} \sigma^{\alpha} (1 - x_{t,i^*}).$$

$\qquad\square$

Then we bound the sum of Bregman divergence in stochastic case by the stopping time argument.

**Theorem 21** (Formal version of Theorem 8). *In stochastic settings, the sum of Bregman divergence terms can be bounded by*

$$\mathbb{E}\left[\sum_{t=1}^{T} \text{DIV}_t\right] \leq 8192 \cdot 16384^{\frac{1}{\alpha-1}} \cdot K \Delta_{\min}^{-\frac{1}{\alpha-1}} \sigma^{\frac{\alpha}{\alpha-1}} \log T + \frac{1}{4} \mathcal{R}_T.$$

*Proof.* We first consider a stopping threshold $M_1$

$$M_1 := \inf\left\{s > 0 : 4096 \cdot s^{1-\alpha}\sigma^\alpha \leq \Delta_{\min}/4\right\} \tag{35}$$

Then, define the stopping time $\mathcal{T}_1$ as

$$\mathcal{T}_1 := \inf\{t \geq 1 : S_t \geq M_1\} \wedge (T+1).$$

Since we have $S_{t+1} \leq (1 + (4K\log T)^{-1})S_t \leq 2S_t$ by Eq. (30), we have $S_{\mathcal{T}_1} \leq 2M_1$. Then by Lemma 17, we have

$$\mathbb{E}\left[\sum_{t=1}^{\mathcal{T}-1}\text{DIV}_t\right] \leq 4096 \cdot \mathbb{E}\left[S_{\mathcal{T}_1} \cdot (K\log T)\right] \leq 8192 \cdot M_1 \cdot K\log T.$$

Therefore, by Lemma 20,

$$\mathbb{E}\left[\sum_{t=1}^{T}\text{DIV}_t\right] = \mathbb{E}\left[\sum_{t=1}^{\mathcal{T}_1-1}\text{DIV}_t\right] + \mathbb{E}\left[\sum_{t=\mathcal{T}_1}^{T}\mathbb{E}[\text{DIV}_t \mid \mathcal{F}_{t-1}]\right]$$

$$\leq 8192 \cdot M_1 \cdot K\log T + \mathbb{E}\left[\sum_{t=\mathcal{T}_1}^{T}4096 \cdot \mathbb{E}[\sigma^\alpha S_t^{1-\alpha}(1-x_{t,i^*}) \mid \mathcal{F}_{t-1}]\right]$$

$$\leq 8192 \cdot M_1 \cdot K\log T + \mathbb{E}\left[\sum_{t=\mathcal{T}_1}^{T}4096 \cdot M_1^{1-\alpha}\sigma^\alpha(1-x_{t,i^*})\right],$$

where the last inequality is due to $S_t \geq M$ for $t \geq \mathcal{T}_1$ and $1 - \alpha < 0$. Therefore, by definition of $M_1$, we have

$$\mathbb{E}\left[\sum_{t=1}^{T}\text{DIV}_t\right] \leq 8192 \cdot \frac{\Delta_{\min}^{-\frac{1}{\alpha-1}}}{(4 \cdot 4096)^{-\frac{1}{\alpha-1}}} \cdot \sigma^{\frac{\alpha}{\alpha-1}} \cdot K\log T + \frac{\Delta_{\min}}{4}\mathbb{E}\left[\sum_{t=\mathcal{T}_1}^{T}(1-x_{t,i^*})\right]$$

$$\leq 8192 \cdot 16384^{\frac{1}{\alpha-1}}\Delta_{\min}^{-\frac{1}{\alpha-1}} \cdot \sigma^{\frac{\alpha}{\alpha-1}} \cdot K\log T + \frac{1}{4}\mathcal{R}_T,$$

where the first inequality is due to the definition of $M_1$ in Eq. (35) and the second inequality is due to $\mathcal{R}_T \geq \mathbb{E}[\Delta_{\min}\sum_{t=1}^{T}1-x_{t,i^*}]$ in stochastic case. $\qquad\square$

## D $\quad\Psi$-Shifting Terms: Omitted Proofs in Section 5.3

### D.1 $\quad\Psi$-Shifting Terms before Expectation: Proof of Lemma 9

First we give the formal version of Lemma 9.

**Lemma 22** (Formal version of Lemma 9). *We have for any $t \in [T]$,*

$$\text{SHIFT}_t \leq \sum_{i\in[K]}(S_{t+1} - S_t)(-\log(\widetilde{y}_i)) \leq \frac{1}{2}S_t^{-1}(\ell_{t,i_t}^{clip})^2(1-x_{t,i_t})^2.$$

*Moreover, for any $\mathcal{T} \in [T-1]$, we have*

$$\sum_{t=0}^{\mathcal{T}}\text{SHIFT}_t \leq S_{\mathcal{T}+1} \cdot K\log T$$

*Proof.* By definition of $\Psi_t$ in Eq. (2), we have

$$\text{SHIFT}_t = (\Psi_{t+1}(\widetilde{\boldsymbol{y}}) - \Psi_t(\widetilde{\boldsymbol{y}})) - (\Psi_{t+1}(\boldsymbol{x}_{t+1}) - \Psi_t(\boldsymbol{x}_{t+1}))$$

$$\leq \sum_{i\in[K]}(S_{t+1} - S_t)(-\log(\widetilde{y}_i))$$

$$\leq (K\log T) \cdot \frac{S_{t+1}^2 - S_t^2}{S_{t+1} + S_t}$$

$$\leq \frac{1}{2} S_t^{-1} (\ell_{t,i_t}^{\text{clip}})^2 (1 - x_{t,i_t})^2,$$

where the first inequality is by $\Psi_{t+1}(\boldsymbol{x}) \geq \Psi_t(\boldsymbol{x})$, the second inequality is due to the definition of $\widetilde{\boldsymbol{y}}$, and the third inequality holds by $S_{t+1} \geq S_t$. Moreover, for any $\mathcal{T} \in [T - 1]$, we also have

$$\sum_{t=0}^{\mathcal{T}} \text{SHIFT}_t = \sum_{t=0}^{\mathcal{T}} (\Psi_{t+1}(\widetilde{\boldsymbol{y}}) - \Psi_t(\widetilde{\boldsymbol{y}})) - (\Psi_{t+1}(\boldsymbol{x}_t) - \Psi_t(\boldsymbol{x}_t))$$
$$\leq \sum_{t=0}^{\mathcal{T}} (S_{t+1} - S_t) \cdot K \log T$$
$$= S_{\mathcal{T}+1} \cdot K \log T.$$

$\square$

## D.2 Adversarial Bounds for $\Psi$-Shifting Terms: Proof of Theorem 10

In adversarial case, we have already bounded the expectation of $\mathbb{E}[S_{T+1}]$ in Appendix C.5. Therefore, we have the following lemma.

**Theorem 23** (Formal version of Theorem 10). *The expectation of the sum of $\Psi$-shifting terms can be bounded by*

$$\mathbb{E}\left[\sum_{t=0}^{T-1} \text{SHIFT}_t\right] \leq 2 \cdot \sigma K^{1-1/\alpha} T^{1/\alpha} (\log T)^{1-1/\alpha}.$$

*Proof.* By Lemma 22, we have

$$\mathbb{E}\left[\sum_{t=0}^{T-1} \text{SHIFT}_t\right] \leq \mathbb{E}[S_T] \cdot K \log T \leq \mathbb{E}[S_{T+1}] \cdot K \log T.$$

Notice that Lemma 6 gives the bound of $\mathbb{E}[S_{T+1}]$. Therefore, we have

$$\mathbb{E}\left[\sum_{t=0}^{T-1} \text{SHIFT}_t\right] \leq 2 \cdot \sigma K^{1-1/\alpha} T^{1/\alpha} (\log T)^{1-1/\alpha}.$$

$\square$

## D.3 Stochastic Bounds for $\Psi$-Shifting Terms: Proof of Theorem 11

Again, we start with a single-step bound on $\text{SHIFT}_t$.

**Lemma 24.** *Conditioning on $\mathcal{F}_{t-1}$ for any $t \in [T - 1]$, we have*

$$\mathbb{E}[\text{SHIFT}_t \mid \mathcal{F}_{t-1}] \leq S_t^{1-\alpha} \sigma^\alpha (1 - x_{t,i^*})$$

*Proof.* According to Lemma 22, we have

$$\mathbb{E}[\text{SHIFT}_t \mid \mathcal{F}_{t-1}] \leq \frac{1}{2} S_t^{-1} \mathbb{E}\left[\ell_{t,i_t}^{\text{clip}} (1 - x_{t,i_t})^2 \Big| \mathcal{F}_{t-1}\right].$$

By definition of $C_{t,i_t}$ in Eq. (3), we have

$$\mathbb{E}[\text{SHIFT}_t \mid \mathcal{F}_{t-1}] \leq \frac{1}{2} S_t^{-1} \cdot \mathbb{E}\left[C_{t,i_t}^{2-\alpha} (1 - x_{t,i_t})^{2-\alpha} |\ell_{t,i_t}|^\alpha (1 - x_{t,i_t})^\alpha \mid \mathcal{F}_{t-1}\right]$$
$$\leq \frac{S_t^{1-\alpha}}{2 \cdot 4^{2-\alpha}} \sigma^\alpha \mathbb{E}[(1 - x_{t,i_t})^\alpha \mid \mathcal{F}_{t-1}]$$
$$\leq \frac{1}{2} S_t^{1-\alpha} \sigma^\alpha \sum_{i \in [K]} x_{t,i} (1 - x_{t,i})^\alpha$$

By similar method used in the proof of Lemma 20, we further have

$$\sum_{i \in [K]} x_{t,i}(1 - x_{t,i})^\alpha \leq \sum_{i \neq i^*} x_{t,i} + x_{t,i^*}(1 - x_{t,i^*}) \leq 2(1 - x_{t,i^*}).$$

Therefore, we have

$$\mathbb{E}[\text{SHIFT}_t \mid \mathcal{F}_{t-1}] \leq S_t^{1-\alpha}\sigma^\alpha(1 - x_{t,i^*})$$

$\square$

**Theorem 25** (Formal version of Theorem 11). *We can bound the sum of $\Psi$-shifting terms by the following inequality.*

$$\mathbb{E}\left[\sum_{t=0}^{T-1} \text{SHIFT}_t\right] \leq 2K\sigma^{\frac{\alpha}{\alpha-1}}\Delta_{\min}^{-\frac{1}{\alpha-1}}\log T + \frac{1}{4}\mathcal{R}_T$$

*Proof.* Similar to the proof in Theorem 21, we consider a stopping threshold $M_2$ defined as follows

$$M_2 := \inf\left\{s > 0 : s^{1-\alpha}\sigma^\alpha \leq \Delta_{\min}/4\right\}.$$

Then we similarly define the stopping time $\mathcal{T}_2$ as

$$\mathcal{T}_2 := \inf\left\{t \geq 1 : S_t \geq M_2\right\} \wedge T.$$

Therefore, we have $S_{\mathcal{T}} \leq 2S_{\mathcal{T}-1} \leq 2M_2$. Then by Lemma 22, we have

$$\mathbb{E}\left[\sum_{t=0}^{\mathcal{T}_2} \text{SHIFT}_t\right] \leq \mathbb{E}[S_{\mathcal{T}_2+1}] \cdot K\log T \leq 2M_2 \cdot K\log T$$

Therefore, we have

$$\mathbb{E}\left[\sum_{t=0}^{T-1} \text{SHIFT}_t\right] = \mathbb{E}\left[\sum_{t=0}^{\mathcal{T}_2-1} \text{SHIFT}_t\right] + \mathbb{E}\left[\sum_{t=\mathcal{T}_2}^{T-1} \mathbb{E}[\text{SHIFT}_t \mid \mathcal{F}_{t-1}]\right]$$

$$\leq 2M_2 \cdot K\log T + \mathbb{E}\left[\sum_{t=\mathcal{T}_2}^{T-1} \sigma^\alpha \mathbb{E}[S_t^{1-\alpha}(1 - x_{t,i^*}) \mid \mathcal{F}_{t-1}]\right]$$

$$\leq 2M_2 \cdot K\log T + \mathbb{E}\left[\sum_{t=\mathcal{T}_2}^{T-1} \sigma^\alpha M_2^{1-\alpha}\mathbb{E}[(1 - x_{t,i^*}) \mid \mathcal{F}_{t-1}]\right]$$

where the first inequality holds by Lemma 24 and the second inequality is due to $S_t \geq M_2$ for $t \geq \mathcal{T}_2$ and $1 - \alpha < 0$. Notice that $\mathcal{R}_T \geq \mathbb{E}[\sum_{t=1}^{T} \Delta_{\min}(1 - x_{t,i^*})]$. We can combine the definition of $M_2$ and get

$$\mathbb{E}\left[\sum_{t=0}^{T-1} \text{SHIFT}_t\right] \leq 2 \cdot \Delta_{\min}^{-\frac{1}{\alpha-1}}\sigma^{\frac{\alpha}{\alpha-1}} \cdot K\log T + \frac{1}{4}\mathcal{R}_T$$

$\square$

# E  SKIPPING LOSS TERMS: OMITTED PROOFS IN SECTION 5.4

## E.1  CONSTANT CLIPPING THRESHOLD ARGUMENT: PROOF OF LEMMA 12

*Proof of Lemma 12.* For a given clipping threshold constant $M$, we first perform clipping operation $\text{Clip}(\ell_{t,i_t}, M)$, which gives a *universal* clipping error

$$\text{SKIPERR}_t^{\text{Univ}}(M) := \begin{cases} \ell_{t,i_t} - M & \ell_{t,i_t} > M \\ 0 & -M \leq \ell_{t,i_t} \leq M \\ \ell_{t,i_t} + M & \ell_{t,i_t} < -M \end{cases}.$$

If $C_{t,i_t} \geq M$, then this clipping is ineffective. Otherwise, we get the following *action-dependent* clipping error

$$\text{SKIPERR}_t^{\text{ActDep}}(M) := \begin{cases} \text{SKIPERR}_t - \text{SKIPERR}_t^{\text{Univ}} & C_{t,i_t} < M \\ 0 & C_{t,i_t} \geq M \end{cases}.$$

Therefore, we directly have $|\text{SKIPERR}_t| \leq |\text{SKIPERR}_t^{\text{Univ}}(M)| + |\text{SKIPERR}_t^{\text{ActDep}}(M)|$. Another important observation is that if $\text{SKIPERR}_t^{\text{ActDep}}(M) \neq 0$, we have $S_t \leq M$, $(\ell_{t,i_t}^{\text{clip}})^2 = C_{t,i_t}^2$, and $|\text{SKIPERR}_t^{\text{ActDep}}(M)| \leq M$, which implies that

$$S_{t+1}^2 = S_t^2 + C_{t,i_t}^2 \cdot (1 - x_{t,i_t})^2 \cdot (K \log T)^{-1} = S_t^2 \left(1 + (16K \log T)^{-1}\right).$$

Thus the number of occurrences of non-zero $\text{SKIPERR}_t^{\text{ActDep}}(M)$'s is upper-bounded by $\lceil \log_{\sqrt{1+(K \log T)^{-1}/4}} M \rceil$. Then we can bound the sum of the sub-optimal skipping loss by

$$\mathbb{E}\left[\sum_{t=1}^{T} |\text{SKIPERR}_t| \cdot \mathbb{1}[i_t \neq i^*]\right]$$

$$\leq \mathbb{E}\left[\sum_{t=1}^{T} |\text{SKIPERR}_t^{\text{Univ}}(M)| \cdot \mathbb{1}[i_t \neq i^*]\right] + \mathbb{E}\left[\sum_{t=1}^{T} |\text{SKIPERR}_t^{\text{ActDep}}(M)|\right]$$

$$\leq \mathbb{E}\left[\sum_{t=1}^{T} |\ell_{t,i_t}| \cdot \mathbb{1}[|\ell_{t,i_t} \geq M] \cdot \mathbb{1}[i_t \neq i^*]\right] + M \cdot \mathbb{E}[\log_{\sqrt{1+(16K \log T)^{-1}}} M] + M$$

$$\leq \sigma^\alpha M^{1-\alpha} \mathbb{E}\left[\sum_{t=1}^{T} \mathbb{1}[i_t \neq i^*]\right] + 2M \cdot \frac{\log M}{\log(1 + (16K \log T)^{-1})} + M.$$

$\square$

## E.2 Adversarial Bounds for Skipping Losses: Proof of Theorem 13

**Theorem 26** (Formal version of Theorem 13). *By setting $M^{adv} := \sigma(K \log T)^{-1/\alpha} T^{1/\alpha}$, we have*

$$\mathbb{E}\left[\sum_{t=1}^{T} |\text{SKIPERR}_t| \cdot \mathbb{1}[i_t \neq i^*]\right]$$

$$\leq \sigma K^{1-1/\alpha} T^\alpha \cdot \left((\log T)^{1-1/\alpha} + \frac{2}{\alpha}(\log T)^{2-1/\alpha} + 2\log \sigma - \frac{2}{\alpha}\log K - \frac{2}{\alpha}\log \log T\right)$$

*Proof.* Notice that we have $2x \geq \log(1 + x^{-1})$ holding for $x \geq 1$. Therefore,

$$\mathbb{E}\left[\sum_{t=1}^{T} |\text{SKIPERR}_t \cdot \mathbb{1}[i_t \neq i^*]|\right]$$

$$\leq \sigma^\alpha (M^{adv})^{1-\alpha} \mathbb{E}\left[\sum_{t=1}^{T} \mathbb{1}[i_t \neq i^*]\right] + 2M^{adv} \cdot \log M^{adv} \cdot K \log T.$$

By the expression of $M^{adv}$, we have

$$\mathbb{E}\left[\sum_{t=1}^{T} |\text{SKIPERR}_t \cdot \mathbb{1}[i_t \neq i^*]|\right]$$

$$\leq \sigma(K \log T)^{1-1/\alpha} T^{1/\alpha} + 16 \cdot \sigma(K \log T)^{1-1/\alpha} T^{1/\alpha} \cdot \log\left(\sigma(K \log T)^{-1/\alpha} T^{1/\alpha}\right)$$

$$= \sigma K^{1-1/\alpha} T^{1/\alpha} \cdot \left((\log T)^{1-1/\alpha} + \frac{2}{\alpha}(\log T)^{2-1/\alpha} + 2\log \sigma - \frac{2}{\alpha}\log K - \frac{2}{\alpha}\log \log T\right).$$

$\square$

### E.3 STOCHASTIC BOUNDS FOR SKIPPING LOSSES: PROOF OF THEOREM 14

**Theorem 27** (Formal version of Theorem 14). *By setting $M^{sto} := 4^{\frac{1}{\alpha-1}} \sigma^{\frac{\alpha}{\alpha-1}} \Delta_{\min}^{-\frac{1}{\alpha-1}}$, we have*

$$\mathbb{E}\left[\sum_{t=1}^{T} |\text{SKIPERR}_t| \cdot \mathbb{1}[i_t \neq i^*]\right]$$

$$\leq \sigma^{\frac{\alpha}{\alpha-1}} \Delta_{\min}^{-\frac{1}{\alpha-1}} \cdot K \log T \cdot \frac{4^{\frac{1}{\alpha-1}}}{\alpha-1} \left(2 \log 4 + \alpha \log \sigma + \log(1/\Delta_{\min})\right) + \frac{1}{4} \mathcal{R}_T.$$

*Proof.* Since we set the constant in stochastic case as $M^{sto} := 4^{\frac{1}{\alpha-1}} \sigma^{\frac{\alpha}{\alpha-1}} \Delta_{\min}^{-\frac{1}{\alpha-1}}$, then by $2x \geq \log(1 + x^{-1}), \forall x \geq 1$, we have

$$\mathbb{E}\left[\sum_{t=1}^{T} |\text{SKIPERR}_t \cdot \mathbb{1}[i_t \neq i^*]|\right]$$

$$\leq \sigma^{\alpha}(M^{sto})^{1-\alpha} \mathbb{E}\left[\sum_{t=1}^{T} \mathbb{1}[i_t \neq i^*]\right] + 2M^{sto} \cdot \log M^{sto} \cdot K \log T$$

$$= \frac{1}{4}\Delta_{\min}\mathbb{E}\left[\sum_{t=1}^{T} \mathbb{1}[i_t \neq i^*]\right] + \sigma^{\frac{\alpha}{\alpha-1}} \Delta_{\min}^{-\frac{1}{\alpha-1}} \cdot K \log T \cdot \frac{4^{\frac{1}{\alpha-1}}}{\alpha-1} \left(2 \log 4 + \alpha \log \sigma + \log(1/\Delta_{\min})\right)$$

Since we have

$$\mathcal{R}_T \geq \mathbb{E}\left[\sum_{t=1}^{T} \Delta_{\min} \mathbb{1}[i_t \neq i^*]\right],$$

which shows that

$$\mathbb{E}\left[\sum_{t=1}^{T} |\text{SKIPERR}_t \cdot \mathbb{1}[i_t \neq i^*]|\right]$$

$$\leq \sigma^{\frac{\alpha}{\alpha-1}} \Delta_{\min}^{-\frac{1}{\alpha-1}} \cdot K \log T \cdot \frac{4^{\frac{1}{\alpha-1}}}{\alpha-1} \left(2 \log 4 + \alpha \log \sigma + \log(1/\Delta_{\min})\right) + \frac{1}{4} \mathcal{R}_T.$$

$\square$

## F MAIN THEOREM: PROOF OF THEOREM 3

In this section, we prove the main theorem. Here we present the formal version of our main result.

**Theorem 28** (Formal version of Theorem 3). *For Algorithm 1, under the adversarial settings, we have*

$$\mathcal{R}_T \leq \sigma K^{1-1/\alpha} T^{\alpha} \cdot \left(8195(\log T)^{1-1/\alpha} + \frac{2}{\alpha}(\log T)^{2-1/\alpha} + 2 \log \sigma - \frac{2}{\alpha} \log K - \frac{2}{\alpha} \log \log T\right) + \sigma K$$

$$= \widetilde{\mathcal{O}}\left(\sigma K^{1-1/\alpha} T^{1/\alpha}\right).$$

*Moreover, for the stochastic settings, we have*

$$\mathcal{R}_T \leq 4 \cdot K \left(\frac{\sigma^{\alpha}}{\Delta_{\min}}\right)^{\frac{1}{\alpha-1}} \log T \cdot \left(8192 \cdot 16384^{\frac{1}{\alpha-1}} + 2 + \frac{4^{\frac{1}{\alpha-1}}}{\alpha-1}\left(2 \log 4 + \log\left(\frac{\sigma^{\alpha}}{\Delta_{\min}}\right)\right)\right)$$

$$= \mathcal{O}\left(K\left(\frac{\sigma^{\alpha}}{\Delta_{\min}}\right)^{\frac{1}{\alpha-1}} \log T \cdot \log \frac{\sigma^{\alpha}}{\Delta_{\min}}\right).$$

*Proof.* We denote $\boldsymbol{y} \in \mathbb{R}^K$ as the one-hot vector on the optimal action $i^* \in [K]$, *i.e.*, $y_i := \mathbb{1}[i = i^*]$. By definition of $\mathcal{R}_T$ in Eq. (1), we have

$$\mathcal{R}_T = \mathbb{E}\left[\sum_{t=1}^{T} \langle \boldsymbol{x}_t - \boldsymbol{y}, \boldsymbol{\ell}_t \rangle\right].$$

Consider the adjusted benchmark $\widetilde{\boldsymbol{y}}$ where

$$\widetilde{y}_i := \begin{cases} \frac{1}{T} & i \neq i^* \\ 1 - \frac{K-1}{T} & i = i^* \end{cases}.$$

By standard regret decomposition in FTRL-based MAB algorithm analyses, we have

$$\mathcal{R}_T = \underbrace{\mathbb{E}\left[\sum_{t=1}^{T} \langle \widetilde{\boldsymbol{y}} - \boldsymbol{y}, \boldsymbol{\ell}_t^{\text{skip}} \rangle\right]}_{\text{Benchmark Calibration Error}} + \underbrace{\mathbb{E}\left[\sum_{t=1}^{T} \langle \boldsymbol{x}_t - \widetilde{\boldsymbol{y}}, \boldsymbol{\ell}_t^{\text{skip}} \rangle\right]}_{\text{Main Regret}} + \underbrace{\mathbb{E}\left[\sum_{t=1}^{T} \langle \boldsymbol{x}_t - \boldsymbol{y}, \boldsymbol{\ell}_t - \boldsymbol{\ell}_t^{\text{skip}} \rangle\right]}_{\text{Skipping Error}}. \quad (36)$$

As in a typical log-barrier analysis, the Benchmark Calibration Error is not the dominant term. This is because we have, by definitions of $\boldsymbol{y}$ and $\widetilde{\boldsymbol{y}}$,

$$\mathbb{E}\left[\sum_{t=1}^{T} \langle \widetilde{\boldsymbol{y}} - \boldsymbol{y}, \boldsymbol{\ell}_t^{\text{skip}} \rangle\right] \leq \sum_{t=1}^{T} \frac{K-1}{T} \mathbb{E}[|\ell_{t,i_t}^{\text{skip}}|] \leq \sum_{t=1}^{T} \frac{K-1}{T} \mathbb{E}[|\ell_{t,i_t}|] \leq \sigma K,$$

which is independent from $T$. Therefore, the key is analyzing the other two terms.

By the FTRL decomposition in Lemma 29, we have

$$\text{Main Regret} = \mathbb{E}\left[\sum_{t=1}^{T} \langle \boldsymbol{x}_t - \widetilde{\boldsymbol{y}}, \boldsymbol{\ell}_t^{\text{skip}} \rangle\right] = \mathbb{E}\left[\sum_{t=1}^{T} \langle \boldsymbol{x}_t - \widetilde{\boldsymbol{y}}, \widetilde{\boldsymbol{\ell}}_t \rangle\right]$$

$$\leq \sum_{t=1}^{T} \mathbb{E}[D_{\Psi_t}(\boldsymbol{x}_t, \boldsymbol{z}_t)] + \sum_{t=0}^{T-1} \mathbb{E}\left[(\Psi_{t+1}(\widetilde{\boldsymbol{y}}) - \Psi_t(\widetilde{\boldsymbol{y}})) - (\Psi_{t+1}(\boldsymbol{x}_{t+1}) - \Psi_t(\boldsymbol{x}_{t+1}))\right],$$

where

$$D_{\Psi_t}(\boldsymbol{y}, \boldsymbol{x}) = \Psi_t(\boldsymbol{y}) - \Psi_t(\boldsymbol{x}) - \langle \nabla \Psi_t(\boldsymbol{x}), \boldsymbol{y} - \boldsymbol{x} \rangle,$$

given the Bregman divergence induced by the $t$-th regularizer $\Psi_t$, and $\boldsymbol{z}_t$ denotes the posterior optimal estimation in episode $t$, namely

$$\boldsymbol{z}_t := \underset{\boldsymbol{z} \in \Delta^{[K]}}{\arg\min} \left(\sum_{s=1}^{t} \langle \widetilde{\boldsymbol{\ell}}_s, \boldsymbol{z} \rangle + \Psi_t(\boldsymbol{z})\right).$$

As mentioned in Section 5, we denote $\text{Div}_t := D_{\Psi_t}(\boldsymbol{x}_t, \boldsymbol{z}_t)$ for the Bregman divergence between $\boldsymbol{x}_t$ and $\boldsymbol{z}_t$ under regularizer $\Psi_t$, $\text{Shift}_t := [(\Psi_{t+1}(\widetilde{\boldsymbol{y}}) - \Psi_t(\widetilde{\boldsymbol{y}})) - (\Psi_{t+1}(\boldsymbol{x}_{t+1}) - \Psi_t(\boldsymbol{x}_{t+1}))]$ be the $\Psi$-shifting term, and $\text{SkipErr}_t := \ell_{t,i_t} - \ell_{t,i_t}^{\text{skip}} = \ell_{t,i_t} \mathbb{1}[|\ell_{t,i_t}| \geq C_{t,i_t}]$ be the sub-optimal skipping losses. Then, we can reduce the analysis of main regret to bounding the sum of Bregman divergence term $\mathbb{E}[\text{Div}_t]$ and $\Psi$-shifting term $\mathbb{E}[\text{Shift}_t]$. Moreover, for sub-optimal skipping losses, we have

$$\langle \boldsymbol{x}_t - \boldsymbol{y}, \boldsymbol{\ell}_t - \boldsymbol{\ell}_t^{\text{skip}} \rangle = \sum_{i \in [K]} (x_{t,i} - y_i) \cdot (\ell_{t,i} - \ell_{t,i}^{\text{skip}})$$

$$\leq \sum_{i \neq i^*} x_{t,i} \cdot \left|\ell_{t,i} - \ell_{t,i}^{\text{skip}}\right| + (x_{t,i^*} - 1) \cdot \left(\ell_{t,i^*} - \ell_{t,i^*}^{\text{skip}}\right)$$

$$= \mathbb{E}\left[|\text{SkipErr}_t| \cdot \mathbb{1}[i_t \neq i^*] \mid \mathcal{F}_{t-1}\right] + (x_{t,i^*} - 1) \cdot \left(\ell_{t,i^*} - \ell_{t,i^*}^{\text{skip}}\right).$$

Notice that the factor $(x_{t,i^*} - 1)$ in the second term is negative and $\mathcal{F}_{t-1}$-measurable. Then we have

$$\mathbb{E}\left[\ell_{t,i^*} - \ell_{t,i^*}^{\text{skip}} \Big| \mathcal{F}_{t-1}\right] = \mathbb{E}\left[\mathbb{1}[|\ell_{t,i^*}| \geq C_{t,i^*}] \cdot \ell_{t,i^*}\right] \geq 0,$$

where the inequality is due to the truncated non-negative assumption (Assumption 1) of the optimal arm $i^*$. Therefore, we have $\mathbb{E}[(x_{t,i^*} - 1) \cdot (\ell_{t,i^*} - \ell_{t,i^*}^{\text{skip}}) \mid \mathcal{F}_{t-1}] \leq 0$ and thus

$$\mathbb{E}[\langle \boldsymbol{x}_t - \boldsymbol{y}, \boldsymbol{\ell}_t - \boldsymbol{\ell}_t^{\text{skip}} \rangle \mid \mathcal{F}_{t-1}] \leq \mathbb{E}[|\text{SKIPERR}_t| \cdot \mathbb{1}[i_t \neq i^*] \mid \mathcal{F}_{t-1}],$$

which gives an approach to control the skipping error by the sum of skipping losses $\text{SKIPERR}_t$'s where we pick a sub-optimal arm $i_t \neq i^*$. Formally, we give the following inequality:

$$\text{SKIPPING ERROR} \leq \mathbb{E}\left[\sum_{t=1}^{T} |\text{SKIPERR}_t| \cdot \mathbb{1}[i_t \neq i^*]\right].$$

To summarize, the regret $\mathcal{R}_T$ decomposes into the sum of Bregman divergence terms $\mathbb{E}[\text{DIV}_t]$, the $\Psi$-shifting terms $\mathbb{E}[\text{SHIFT}_t]$, and the sub-optimal skipping losses $\mathbb{E}[|\text{SKIPERR}_t| \cdot \mathbb{1}[i_t \neq i^*]]$, namely

$$\mathcal{R}_T \leq \underbrace{\mathbb{E}\left[\sum_{t=1}^{T} \text{DIV}_t\right]}_{\text{BREGMAN DIVERGENCE TERMS}} + \underbrace{\mathbb{E}\left[\sum_{t=0}^{T-1} \text{SHIFT}_t\right]}_{\Psi\text{-SHIFTING TERMS}} + \underbrace{\mathbb{E}\left[\sum_{t=1}^{T} |\text{SKIPERR}_t| \cdot \mathbb{1}[i_t \neq i^*]\right]}_{\text{SUB-OPTIMAL SKIPPING LOSSES}} + \sigma K.$$

We discuss the regret upper bound under adversarial and stochastic environments separately.

**Adversarial Cases.** According to Theorems 19, 23, and 26, we have

$$\mathcal{R}_T \leq \mathbb{E}\left[\sum_{t=1}^{T} \text{DIV}_t\right] + \mathbb{E}\left[\sum_{t=0}^{T-1} \text{SHIFT}_t\right] + \mathbb{E}\left[\sum_{t=1}^{T} |\text{SKIPERR}_t| \cdot \mathbb{1}[i_t \neq i^*]\right] + \sigma K$$

$$\leq 8192 \cdot \sigma K^{1-1/\alpha} T^{1/\alpha} (\log T)^{1-1/\alpha} + 2 \cdot \sigma K^{1-1/\alpha} T^{1/\alpha} (\log T)^{1-1/\alpha}$$

$$+ \sigma K^{1-1/\alpha} T^{1/\alpha} \cdot \left((\log T)^{1-1/\alpha} + \frac{2}{\alpha}(\log T)^{2-1/\alpha} + 2\log\sigma - \frac{2}{\alpha}\log K - \frac{2}{\alpha}\log\log T\right) + \sigma K$$

$$= \sigma K^{1-1/\alpha} T^{1/\alpha} \cdot \left(8195(\log T)^{1-1/\alpha} + \frac{2}{\alpha}(\log T)^{2-1/\alpha} + 2\log\sigma - \frac{2}{\alpha}\log K - \frac{2}{\alpha}\log\log T\right) + \sigma K.$$

**Stochastic Cases.** According to Theorems 21, 25, and 27, we have

$$\mathcal{R}_T \leq \mathbb{E}\left[\sum_{t=1}^{T} \text{DIV}_t\right] + \mathbb{E}\left[\sum_{t=0}^{T-1} \text{SHIFT}_t\right] + \mathbb{E}\left[\sum_{t=1}^{T} |\text{SKIPERR}_t| \cdot \mathbb{1}[i_t \neq i^*]\right] + \sigma K$$

$$\leq 8192 \cdot 16384^{\frac{1}{\alpha-1}} \cdot K \Delta_{\min}^{-\frac{1}{\alpha-1}} \sigma^{\frac{\alpha}{\alpha-1}} \log T + \frac{1}{4}\mathcal{R}_T$$

$$+ 2K\sigma^{\frac{\alpha}{\alpha-1}} \Delta_{\min}^{-\frac{1}{\alpha-1}} \log T + \frac{1}{4}\mathcal{R}_T$$

$$+ \sigma^{\frac{\alpha}{\alpha-1}} \Delta_{\min}^{-\frac{1}{\alpha-1}} \cdot K \log T \cdot \frac{4^{\frac{1}{\alpha-1}}}{\alpha-1} \left(2\log 4 + \alpha\log\sigma + \log(1/\Delta_{\min})\right) + \frac{1}{4}\mathcal{R}_T + \sigma K$$

$$\leq K\left(\frac{\sigma^\alpha}{\Delta_{\min}}\right)^{\frac{1}{\alpha-1}} \log T \cdot \left(8192 \cdot 16384^{\frac{1}{\alpha-1}} + 2 + \frac{4^{\frac{1}{\alpha-1}}}{\alpha-1}\left(2\log 4 + \log\left(\frac{\sigma^\alpha}{\Delta_{\min}}\right)\right)\right) + \frac{3}{4}\mathcal{R}_T,$$

which implies that

$$\mathcal{R}_T \leq 4 \cdot K\left(\frac{\sigma^\alpha}{\Delta_{\min}}\right)^{\frac{1}{\alpha-1}} \log T \cdot \left(8192 \cdot 16384^{\frac{1}{\alpha-1}} + 2 + \frac{4^{\frac{1}{\alpha-1}}}{\alpha-1}\left(2\log 4 + \log\left(\frac{\sigma^\alpha}{\Delta_{\min}}\right)\right)\right).$$

Therefore, we finish the proof, which shows the BoBW property of Algorithm 1. $\qquad\square$

The following lemma characterizes the FTRL regret decomposition, which is the extension of the classical FTRL bound (Lattimore & Szepesvári, 2020, Theorem 28.5). Dann et al. (2023a, Lemma 17) also gave a similar result, but we include a full proof here for the sake of completeness.

**Lemma 29** (FTRL Regret Decomposition). *In* `uniINF` *(Algorithm [1]), we have (set $S_0 = 0$ for simplicity),*

$$\mathbb{E}\left[\sum_{t=1}^{T}\langle \boldsymbol{x}_t - \widetilde{\boldsymbol{y}}, \widetilde{\boldsymbol{\ell}}_t\rangle\right] \le \sum_{t=1}^{T}\mathbb{E}[D_{\Psi_t}(\boldsymbol{x}_t, \boldsymbol{z}_t)] + \mathbb{E}[(\Psi_t(\widetilde{\boldsymbol{y}}) - \Psi_{t-1}(\widetilde{\boldsymbol{y}})) - (\Psi_t(\boldsymbol{x}_t) - \Psi_{t-1}(\boldsymbol{x}_t))],$$

*where $D_{\Psi_t}$ is the Bregman divergence induced by $\Psi_t$, and $\boldsymbol{z}_t$ is given by*

$$\boldsymbol{z}_t := \operatorname*{argmin}_{\boldsymbol{z}\in\Delta^{[K]}}\left(\sum_{s=1}^{t}\langle \widetilde{\boldsymbol{\ell}}_s, \boldsymbol{z}\rangle + \Psi_t(\boldsymbol{z})\right).$$

*Proof.* We denote $\boldsymbol{L}_t := \sum_{s=1}^{t}\widetilde{\boldsymbol{\ell}}_s$. Denote $f^* : \mathbb{R}^k \to \mathbb{R}$ as the Frenchel conjugate of function $f : \mathbb{R}^K \to \mathbb{R}$, where

$$f^*(\boldsymbol{y}) := \sup_{\boldsymbol{x}\in\mathbb{R}^K}\left\{\langle \boldsymbol{y}, \boldsymbol{x}\rangle - f(\boldsymbol{x})\right\}.$$

Moreover, denote $\overline{f} : \mathbb{R}^K \to \mathbb{R}$ as the restriction of $f : \mathbb{R}^K \to \mathbb{R}$ on $\Delta^{[K]}$, i.e,

$$\overline{f}(\boldsymbol{x}) = \begin{cases} f(\boldsymbol{x}), & \boldsymbol{x} \in \Delta^{[K]} \\ \infty, & \boldsymbol{x} \notin \Delta^{[K]} \end{cases}.$$

Therefore, by definition, we have

$$\boldsymbol{z}_t = \nabla\overline{\Psi}_t^*(-\boldsymbol{L}_t), \quad \boldsymbol{x}_t = \nabla\overline{\Psi}_t^*(-\boldsymbol{L}_{t-1}).$$

Then recall the properties of Bregman divergence, we have

$$D_{\Psi_t}(\boldsymbol{x}_t, \boldsymbol{z}_t) = D_{\Psi_t}(\nabla\overline{\Psi}_t^*(-\boldsymbol{L}_{t-1}), \nabla\overline{\Psi}_t^*(-\boldsymbol{L}_t)) = D_{\overline{\Psi}_t^*}(-\boldsymbol{L}_t, -\boldsymbol{L}_{t-1}).$$

Therefore, we have

$$\sum_{t=1}^{T}\langle \boldsymbol{x}_t - \widetilde{\boldsymbol{y}}, \widetilde{\boldsymbol{\ell}}_t\rangle = \langle \widetilde{\boldsymbol{y}}, -\boldsymbol{L}_T\rangle - \sum_{t=1}^{T}\langle \boldsymbol{x}_t, -\widetilde{\boldsymbol{\ell}}_t\rangle$$

$$= \sum_{t=1}^{T}\left(\overline{\Psi}_t^*(-\boldsymbol{L}_t) - \overline{\Psi}_t^*(-\boldsymbol{L}_{t-1}) - \langle\nabla\overline{\Psi}_t^*(-\boldsymbol{L}_{t-1}), -\boldsymbol{L}_t + \boldsymbol{L}_{t-1}\rangle\right)$$

$$\quad + \langle \widetilde{\boldsymbol{y}}, -\boldsymbol{L}_T\rangle - \sum_{t=1}^{T}\left(\overline{\Psi}_t^*(-\boldsymbol{L}_t) - \overline{\Psi}_t^*(-\boldsymbol{L}_{t-1})\right)$$

$$= \sum_{t=1}^{T}D_{\overline{\Psi}_t^*}(-\boldsymbol{L}_t, -\boldsymbol{L}_{t-1}) + \langle \widetilde{\boldsymbol{y}}, -\boldsymbol{L}_T\rangle - \sum_{t=1}^{T}\left(\overline{\Psi}_t^*(-\boldsymbol{L}_t) - \overline{\Psi}_t^*(-\boldsymbol{L}_{t-1})\right)$$

$$= \sum_{t=1}^{T}D_{\Psi_t}(\boldsymbol{x}_t, \boldsymbol{z}_t) + \langle \widetilde{\boldsymbol{y}}, -\boldsymbol{L}_T\rangle - \sum_{t=1}^{T}\left(\overline{\Psi}_t^*(-\boldsymbol{L}_t) - \overline{\Psi}_t^*(-\boldsymbol{L}_{t-1})\right).$$

For the second and third terms, we have

$$\langle \widetilde{\boldsymbol{y}}, -\boldsymbol{L}_T\rangle - \sum_{t=1}^{T}\left(\overline{\Psi}_t^*(-\boldsymbol{L}_t) - \overline{\Psi}_t^*(-\boldsymbol{L}_{t-1})\right)$$

$$= \langle \widetilde{\boldsymbol{y}}, -\boldsymbol{L}_T\rangle - \sum_{t=1}^{T}\left(\overline{\Psi}_t^*(-\boldsymbol{L}_t) - \langle \boldsymbol{x}_t, -\boldsymbol{L}_{t-1}\rangle + \Psi_t(\boldsymbol{x}_t)\right)$$

$$= \langle \widetilde{\boldsymbol{y}}, -\boldsymbol{L}_T\rangle - \sum_{t=1}^{T}\left(\sup_{\boldsymbol{x}\in\Delta^{[K]}}\left\{\langle \boldsymbol{x}, -\boldsymbol{L}_t\rangle - \Psi_t(\boldsymbol{x})\right\} - \langle \boldsymbol{x}_t, -\boldsymbol{L}_{t-1}\rangle + \Psi_t(\boldsymbol{x}_t)\right)$$

$$\le \langle \widetilde{\boldsymbol{y}}, -\boldsymbol{L}_T\rangle - \sum_{t=1}^{T-1}\left(\langle \boldsymbol{x}_{t+1}, -\boldsymbol{L}_t\rangle - \Psi_t(\boldsymbol{x}_{t+1}) - \langle \boldsymbol{x}_t, -\boldsymbol{L}_{t-1}\rangle + \Psi_t(\boldsymbol{x}_t)\right)$$

$$- \sup_{\boldsymbol{x} \in \Delta^{[K]}} \left\{ \langle \boldsymbol{x}, -\boldsymbol{L}_T \rangle - \Psi_T(\boldsymbol{x}) \right\} + \langle \boldsymbol{x}_T, -\boldsymbol{L}_{T-1} \rangle - \Psi_T(\boldsymbol{x}_T)$$

$$= \langle \widetilde{\boldsymbol{y}}, -\boldsymbol{L}_T \rangle - \sum_{t=1}^{T} \left( \Psi_t(\boldsymbol{x}_t) - \Psi_{t-1}(\boldsymbol{x}_t) \right) - \sup_{\boldsymbol{x} \in \Delta^{[K]}} \left\{ \langle \boldsymbol{x}, -\boldsymbol{L}_T \rangle - \Psi_T(\boldsymbol{x}) \right\}$$

$$= \langle \widetilde{\boldsymbol{y}}, -\boldsymbol{L}_T \rangle - \Psi_T(\widetilde{\boldsymbol{y}}) - \sup_{\boldsymbol{x} \in \Delta^{[K]}} \left\{ \langle \boldsymbol{x}, -\boldsymbol{L}_T \rangle - \Psi_T(\boldsymbol{x}) \right\} + \Psi_T(\widetilde{\boldsymbol{y}}) - \sum_{t=1}^{T} \left( \Psi_t(\boldsymbol{x}_t) - \Psi_{t-1}(\boldsymbol{x}_t) \right)$$

$$\leq \sum_{t=1}^{T} \left( \Psi_t(\widetilde{\boldsymbol{y}}) - \Psi_{t-1}(\widetilde{\boldsymbol{y}}) \right) - \sum_{t=1}^{T} \left( \Psi_t(\boldsymbol{x}_t) - \Psi_{t-1}(\boldsymbol{x}_t) \right),$$

which finishes the proof. $\qquad\square$

