# OpenReview forum: "uniINF: Best-of-Both-Worlds Algorithm for Parameter-Free Heavy-Tailed MABs"
_ICLR.cc/2025/Conference — ICLR 2025 Spotlight_

### Official Review · Reviewer_uA58 · 2024-10-19

**Soundness:** 3
**Presentation:** 2
**Contribution:** 3
**Rating:** 8
**Confidence:** 4

**Summary:**

This paper studies parameter-free best-of-both-worlds (BOBW) for HT-MABs, where 1) HT means that the loss distributions can be unbounded but have $\\alpha$-th moment bounded by $\\sigma^{\\alpha}$, for some $\\sigma>0, \\alpha\in(1,2]$; 2) BOBW means that one single algorithm can enjoy logarithmic gap-dependent regret in the stochastic environment (loss distributions are fixed over time) and worst-case optimal regret in adversarial environment (loss distributions change over time), without knowing in advance whether the environment is sto. or not; 3) parameter-free means that the algorithm doesn’t now the value of $\\sigma>0, \\alpha\in(1,2]$, but can ensure the regret guarantee as if they were known.

An algorithm called uniINF is proposed, which ensures $\\tilde{O}(\\frac{K}{(\\Delta_{\\text{min}})^{\\frac{1}{\\alpha-1}}}) $ (expected pseudo-)regret in sto. env. (which is optimal up to log terms and the gap dependency), and near-optimal regret in adv. env. (which is optimal up to log terms) when the loss distributions of the optimal arm satisfy the truncated non-negative assumption (Assumption 1). This is the first parameter-free BOBW result in HTMAB. Previous results approach that in one single env. only (either sto. or adv.).

Technically, this is achieved by several components, including 1) iterative and adaptive learning rate scheduling; 2) adaptive clipping/skipping; 3) refined analysis for log-barrier regularizer.

**Strengths:**

The parameter-free BOBW bound in HTMAB is a quite strong guarantee, and to achieve this, several technical innovations are proposed.

**Weaknesses:**

I’m not happy with how Assumption 1 is justified. From Line 193 to 195, it says that "we make the following **essential** assumption. As shown in (Genalti et al., 2024, Theorems 2 & 3), without Assumption 1, there does not exist HTMAB algorithms that can … without knowing either $\\alpha$ or $\\sigma$." However, this statement could be misleading based on my understanding on (Genalti et al., 2024).

The negative result shown in (Genalti et al., 2024) is that, it’s impossible for one single algorithm to match the lower bound in (Bubeck et al., 2013) for all unknown $\\sigma>0$ or $\\alpha\\in(1,2]$. However, I don’t think it has been characterized that how weak the needed assumption to be "parameter-free" in HTMAB. In fact, in the conclusion part of (Genalti et al., 2024), it even says that "investigating the role of the truncated non-positivity assumption, especially, whether weaker assumptions can be formulated."

Therefore, I would urge the authors to refine the statements related to Assumption 1, as currently it may leave the impression that Assumption 1 is a necessary condition for "parameter-free", which as of now it’s still unclear yet.

**Questions:**

1. To my understanding, this paper claims to develop a new analysis for $DIV_t$ (also named stability term) when using log-barrier, and introduces an extra $(1-x_{t,i})^2$ factor in the bound, which is the key to get self-bounding property and BOBW. However, I don't quite understand what it means by “$S_t$ is adequately large compared to $||c_t||_{\infty}$”. I tried to find a formal lemma or theorem statement for this new bound (with exactly the same form) on $DIV_t$ term but I failed. Could the authors help explain this? Under what conditions does this new bound hold?

2. What’s the difficulty to get the refined gap dependency? From the appendix, I feel that in both DIV term and SHIFT term, we cannot achieve that. Could the authors elaborate more on that (from the analysis perspective)? Is it because the regularizer is log-barrier rather than Tsallis entropy?

Post rebuttal ===============================

I'm increasing the score from 6 to 8.

---

> ### Author Response · Authors · 2024-11-22
> **Author Response**
>
> Thank you very much for your time and effort in reviewing our paper! Please find our responses to your comments below. We will be happy to answer any further questions you may have.
>
> ### Weaknesses
>
> Thank you for pointing out the confusion regarding the truncated non-negativity assumption (Assumption 1) in our work. In light of your feedback, we have revised our discussion on Assumption 1 (see Lines 186-190) and included a clear and detailed summary (see Appendix B).
>
> To be precise, we summarize the exisiting results for the relationship between heavy-tailed MABs and Assumption 1 in the following table.
>
>
> | Assumptions | Known $(\sigma,\alpha)$ | Unknown $(\sigma,\alpha)$ |
> |---|---|---|
> |With Assumption 1 | HTINF achieves BoBW [1] | uniINF achieves BoBW (**This paper**) |
> | Weaker than Assumption 1? | SAO-HT achieves BoBW [2] (see below) | `Open` |
> |Without Any Assumption | SAO-HT achieves BoBW [2] | No BoBW possible, Theorems 2 & 3 in [3] |
>
>
> In particular, as shown in the table,
> - Theorems 2 & 3 in [3] highlight that in parameter-free setups, achieving optimal worst-case regret guarantees is impossible unless further assumptions -- which may or may not be strictly weaker than Assumption 1 -- are made.
> - Our paper demonstrates that, in parameter-free setups, Assumption 1 is sufficient for achieving a BoBW guarantee.
> - Recent work [2] justified that when parameters $(\sigma,\alpha)$ are known, BoBW is achievable without any assumptions (that is, Assumption 1 is redundant when parameters are known).
> - It remains an open question whether a weaker assumption than Assumption 1 could also support BoBW guarantees when $(\sigma, \alpha)$ are unknown.
>
> ### Questions
>
> > Q1: To my understanding, this paper claims to develop a new analysis for $DIV_t$ (also named stability term) when using log-barrier, and introduces an extra $(1 - x_{t,i})^2$ factor in the bound, which is the key to get self-bounding property and BOBW. However, I don't quite understand what it means by “$S_t$ is adequately large compared to $\|c_t\|_\infty$”. I tried to find a formal lemma or theorem statement for this new bound (with exactly the same form) on $DIV_t$ term but I failed. Could the authors help explain this? Under what conditions does this new bound hold?
>
> A: Thanks for your comment. Classically, when bounding the Bregman divergence term $DIV_t$, we require $||c_{t}||\_{\infty} \le O(S_{t})$, resulting in a bound involving the factor $(1 - x_{t,i})$ in the upper bound of $c_t$. This approach is enough to exclude $i^\ast$ in a non-heavy-tail setup, but is insufficient for ours.
> To address this, we have developed a new bounding approach that loosens the magnitude requirement to $|c_{t, i}| \le O(S_t / (1 - x_{t,i}))$ (the exact meaning of "adaquately large") and, at the same time, provides a tighter $DIV_t$ upper-bound with a second-order $(1 - x_{t,i})^2$ factor. For the formal derivation and specific conditions under which this bound holds, we direct readers to Eq. (5) in the main text and the detailed proof provided in Lemma 17.
>
> > Q2: What’s the difficulty to get the refined gap dependency? From the appendix, I feel that in both DIV term and SHIFT term, we cannot achieve that. Could the authors elaborate more on that (from the analysis perspective)? Is it because the regularizer is log-barrier rather than Tsallis entropy?
>
> A: Thank you for sharing your insights on the extra $\log \frac{\sigma^\alpha}{\Delta_{\min}}$ factor in stochastic environments! In fact, the reason of this gap is not log-barrier regularizers (which, indeed causes an extra $\log T$ factor but only in adversarial cases). Instead, it comes from the stopping-time argument for counting the increasement of learning rate, which is sketched in Lines 456-462 (for Div), 486-489 (for Shift), and 502-507 (for skipping), and also detailed in Appendix E.3. Refining the log overhead induced by stopping-time argument is an important future direction.
>
> It is true that Tsallis-entropy regularizers can lead to optimal BoBW results [1]. However, it strongly depend on the prior knowledge of heavy-tailed parameters $(\sigma, \alpha)$, which is unknown in our formulation.
>
> ---
>
> **We hope our responses fully address your concerns. If so, we wonder if you could kindly consider raising your score rating? Meanwhile, we are also more than happy to answer any further questions. Thank you once again for your review!**
>
>
> ### References
>
> [1] Huang, Jiatai, Yan Dai, and Longbo Huang. "Adaptive best-of-both-worlds algorithm for heavy-tailed multi-armed bandits." International Conference on Machine Learning. PMLR, 2022.
>
> [2] Cheng, Duo, Xingyu Zhou, and Bo Ji. "Taming Heavy-Tailed Losses in Adversarial Bandits and the Best-of-Both-Worlds Setting." The Thirty-eighth Annual Conference on Neural Information Processing Systems, 2024.
>
> [3] Genalti, Gianmarco, et al. "$(ε, u) $-Adaptive Regret Minimization in Heavy-Tailed Bandits." The Thirty Seventh Annual Conference on Learning Theory. PMLR, 2024.

---

> > ### Comment · Reviewer_uA58 · 2024-11-22
> >
> > I thank authors for their response. I appreciate their efforts on incorporating reviewers' feedback on Assumption 1 and updating the manuscript.
> >
> > Regarding Q1, I now have a better understanding on the result on DIV term with log-barrier. In terms of Q2, the authors may have messed up and put a response for some other comments here (indeed, to Reviewer jQU5), so the authors may want to reply again regarding my Q2.
> >
> > A minor typo is that in all of the responses, the reference of "HTINF achieves BoBW [x]" shouldn't be correct. The reference should be: Huang, Jiatai, Yan Dai, and Longbo Huang. "Adaptive best-of-both-worlds algorithm for heavy-tailed multi-armed bandits." international conference on machine learning. PMLR, 2022. The authors may want to check if this also happens in the manuscript.

---

> > > ### Author Response · Authors · 2024-11-22
> > >
> > > Thank you very much for reading our response and pointing out our typo! We accidentally mixed the two papers at ICML 2022 and 2023 by Huang et al. when crafting our responses. The manuscript does not contain this mistake. We have corrected all such references in our responses.
> > >
> > > Regarding Q2, we interpreted your question "refined gap dependency" as the $\log \frac{\sigma^\alpha}{\Delta_{\min}}$ term in stochastic cases. In case you were referring to something else, we summarize our sub-optimalities below:
> > > 1. $\log \frac{\sigma^\alpha}{\Delta_{\min}}$ term in stochastic cases: This term arises from our stopping-time argument, which impacts all three components: Div, Shift, and SkipErr. Our perivous response to your Q2 contains more context on this.
> > > 2. $\log T$ term in adversarial cases: This is due to our shift from Tsallis-entropy regularizers to log-barrier regularizers, similar to concurrent work [2] which suffered from a $\log^4 T$ overhead in adversarial cases with log-barrier regularizers.
> > > 3. $\frac{K}{\Delta_{\min}}$ instead of $\sum_{i\ne i^\ast} \frac{1}{\Delta_i}$ in stochastic cases: Technically, to allow gap dependency on every single arm, we shall make the learning rates also arm-dependent. However, in our analysis, it turned out that when excluding the optimal arm, we need to control the increase of cross terms like $S_{t,i}S_{t,j}$ if the arm-dependent learning rate is applied, which is hard to control. Instead, we consider arm-independent learning rate $S_t$ for step $t$ and this cross term becomes $S_t^2$, which made our regret result only adaptive to $\Delta_{\min}$ but not every $\Delta_i$.
> > >
> > > We are really grateful for your detailed feedback. Should this response address your concerns, we would greatly appreciate your consideration in raising your score rating. We are also more than happy to answer any further questions. Thank you once again for your time!

---

> > > > ### Comment · Reviewer_uA58 · 2024-11-22
> > > >
> > > > I thank the authors for the replay.
> > > >
> > > > I meant "refined gap dependency" in my Q2 by the $K/\\Delta_{\\text{min}}$ term. Sorry for not making it clear.
> > > >
> > > > I've also read the communication with other reviewers. This paper overall presents a strong theoretical result in BOBW HTMAB. I'm increasing my score to 8 and I recommend acceptance. I suggest that the authors incorporate (some of) the comments to further improve the presentation in furture versions.

---

> > > > > ### Author Response · Authors · 2024-11-22
> > > > >
> > > > > Thank you very much for your comments! We will definitely incorporate our discussions into our next revision.

---

### Official Review · Reviewer_3K75 · 2024-10-27

**Soundness:** 3
**Presentation:** 3
**Contribution:** 3
**Rating:** 6
**Confidence:** 4

**Summary:**

This work establishes the first parameter-free algorithm for heavy-tailed multi-armed bandits (MABs) with best-of-both-worlds (BOBW) properties. This algorithm does not require prior knowledge of heavy-tail parameters $(\sigma, \alpha)$ and simultaneously obtains the (nearly) optimal regret in both the stochastic and adversarial environments.

**Strengths:**

1. **Technical innovations**: The proposed algorithm and its analysis incorporate new ingredients including a new skipping and clipping scheme of the loss estimates and a stopping time argument to bound the stability terms and the skipping errors, which seem to be technically valuable and might be of independent interest.
2. **Writing**: Generally, this work is well written.

**Weaknesses:**

1. **More comparisons with existing literature**: Most parts of the presentation in this work are clear. However, I would like to suggest the authors provide more discussions and comparisons with the techniques in existing literature. For instance, the exclusion of the optimal arm $i^\ast$ in Eq. (5) when using the log-barrier regularizer is also achieved by [1,2]. I am not very sure whether there are additional technical nuances between the exclusion of the optimal arm in Eq. (5) of this work and those in [1,2]. For the data-dependent learning rates, several works have also leveraged them to achieve BOBW results in various online learning problems (say, [2,3,4,5,6,7]). Besides, when bounding the stability term of OMD/FTRL, a key property required is to ensure the multiplicative stability of the update of the prediction. In this work, such a property is guaranteed by Lemma 4. However, it seems not appropriate to call such a lemma “novel” as on Line 423, since it has also appeared in previous works when using the log-barrier regularizer (say, Lemma 9 in [8]; Lemma 12 in [9]).

[1] Ito. Parameter-Free Multi-Armed Bandit Algorithms with Hybrid Data-Dependent Regret Bounds. COLT, 21.

[2] Ito. Hybrid Regret Bounds for Combinatorial Semi-Bandits and Adversarial Linear Bandits. NeurIPS, 21.

[3] Ito et al. Nearly Optimal Best-of-Both-Worlds Algorithms for Online Learning with Feedback Graphs. NeurIPS, 22.

[4] Tsuchiya et al. Best-of-Both-Worlds Algorithms for Partial Monitoring. ALT, 23.

[5] Ito et al. Best-of-Three-Worlds Linear Bandit Algorithm with Variance-Adaptive Regret Bounds. COLT, 23.

[6] Kong et al. Best-of-three-worlds analysis for linear bandits with follow-the-regularized-leader algorithm. COLT, 23.

[7] Ito et al. Adaptive Learning Rate for Follow-the-Regularized-Leader: Competitive Analysis and Best-of-Both-Worlds. COLT, 24.

[8] Lee et al. A closer look at small-loss bounds for bandits with graph feedback. COLT, 20.

[9] Jin et al. Simultaneously Learning Stochastic and Adversarial Episodic MDPs with Known Transition. NeurIPS, 20.

**Questions:**

1. If $(\sigma, \alpha)$ is known a priori, can we eliminate Assumption 1? What are the main technical difficulties when eliminating Assumption 1 in the case of known $(\sigma, \alpha)$?
2. In the previous work [10], the Tsallis entropy regularizer is used while the log-barrier regularizer is used in this work. Is it because the magnitude of the loss estimates in this work is larger than the magnitude of the loss estimates in [10]?

[10] Huang et al. Adaptive Best-of-Both-Worlds Algorithm for Heavy-Tailed Multi-Armed
Bandits. ICML, 22.

---

> ### Author Response · Authors · 2024-11-22
> **Author Responses (1/2)**
>
> Thank you very much for your time and effort in reviewing our paper! Please find our responses to your comments below. We are more than happy to answer any further questions.
>
> ### Weaknesses
> We appreciate the valuable comments and we include more comparisons in the revision (see Appendix A). Here we answer the three main comments.
> >W1: The exclusion of the optimal arm $i^*$ is also achieved by [1,2]. I am not very sure whether there are additional technical nuance.
>
> A: Thank you for the comment. [1,2] both focus on bandit problems where each action incurs a bounded loss ($[0,1]$-bounded or $[-1,1]$-bounded). For MAB with bounded losses like the setup of [1], it suffices to establish an upper-bound for each $DIV_t$ term with a $(1-x_{t,a_t})$ factor. However, in the heavy-tailed MAB settings, we need an extra power, namely $(1-x_{t,a_t})^2$, as we highlight in Eq. (5).
>
> Technically speaking, when we want to control the second moment of the losses (or establish sharply concentrated mean-loss estimates in UCB-type algorithms like [13]), we need clipping or similar techniques to the losses. In this paper, our clipping threshold contains an $(1 - x_{t,a_t})^{-1}$ factor, which is essential due to other components. Thus, the second moment of the clipped loss contains an $(1-x_{t,a_t})^{1 - \alpha}$ factor, which requires us to bound $DIV_t$ even tighter -- for example our $(1-x_{t,a_t})^2$ term in Eq. (5).
>
> >W2: The related works of data-dependent learning rates:
>
> A: Thank you for your advice. Data-dependent learning rates are very common and crucial in achieving efficient learning in both adversarial and stochastic environments for various MAB problems. We have included more discussion on the data-dependent learning rates in the revision (see Appendix A).
>
> >W3: The multiplicative stability issue:
>
> A: Thank you for pointing out the potential confusion. We did not mean that the statement "$x,z$ multiplicatively close" itself is novel. Our contribution, however, lies in extending this concept to scenarios involving unbounded or heavy-tailed losses, as detailed in Appendix C.1.
>
> For more context, we agree with the reviewer that when losses are $[0,1]$-bounded, previous works such as [8, 9, 11] already showed the multiplicatively close result. However, as we highlight in the text before Eq. (5), existing results on general losses are not yet optimal, which is crucial for heavy-tailed losses. We appreciate your feedback and we have revised our statement (see Lines 420-424).
>
>
>
>
> ### Questions
>
> >Q1:  If $(\sigma, \alpha)$ is known a priori, can we eliminate Assumption 1? What are the main technical difficulties when eliminating Assumption 1 in the case of known $(\sigma, \alpha)$?
>
> A: As indicated in recent research [12], when these parameters are known, Assumption 1 can be eliminated (Please refer to the analysis in [12] for detailed technical insights into how this is accomplished).
>
> However, in scenarios where $(\sigma,\alpha)$ are not known, which we term as the parameter-free setup, the situation is very different. As demonstrated in Theorem 2 & 3 [13], it becomes necessary to implement some assumptions to guide the learning process effectively. These assumptions, while necessary, can potentially be weaker than Assumption 1. Identifying the minimal set of assumptions sufficient for BoBW parameter-free HTMABs forms a crucial ongoing research. We have included more discussion on Assumption 1 in revision (see Appendix B).
>
> >Q2: In the previous work [10], the Tsallis entropy regularizer is used while the log-barrier regularizer is used in this work. Is it because the magnitude of the loss estimates in this work is larger than the magnitude of the loss estimates in [10]?
>
> A: The selection of the log-barrier regularizer in our study over the Tsallis entropy regularizer used in [10] is primarily driven by the different settings we consider. In this paper, we focus on the paremeter-free setting where one does not have access of heavy-tailed parameters $(\sigma, \alpha)$. On the other hand, [10] requires the regularizer to be parametrized by $\alpha$ (as we state in Line 082). Therefore, we opted for the log-barrier regularizer because it does not require prior knowledge of loss distribution parameters and is more suited to the constraints of our parameter-free framework.
>
> ---
>
> **We hope our responses fully address your concerns. If so, we wonder if you could kindly consider raising your score rating? Meanwhile, we are also more than happy to answer any further questions. Thank you once again for your review!**

---

> ### Author Response · Authors · 2024-11-22
> **Author Responses (2/2)**
>
> ### References
>
> [1] Ito. Parameter-Free Multi-Armed Bandit Algorithms with Hybrid Data-Dependent Regret Bounds. COLT, 21.
>
> [2] Ito. Hybrid Regret Bounds for Combinatorial Semi-Bandits and Adversarial Linear Bandits. NeurIPS, 21.
>
> [3] Ito et al. Nearly Optimal Best-of-Both-Worlds Algorithms for Online Learning with Feedback Graphs. NeurIPS, 22.
>
> [4] Tsuchiya et al. Best-of-Both-Worlds Algorithms for Partial Monitoring. ALT, 23.
>
> [5] Ito et al. Best-of-Three-Worlds Linear Bandit Algorithm with Variance-Adaptive Regret Bounds. COLT, 23.
>
> [6] Kong et al. Best-of-three-worlds analysis for linear bandits with follow-the-regularized-leader algorithm. COLT, 23.
>
> [7] Ito et al. Adaptive Learning Rate for Follow-the-Regularized-Leader: Competitive Analysis and Best-of-Both-Worlds. COLT, 24.
>
> [8] Lee et al. A closer look at small-loss bounds for bandits with graph feedback. COLT, 20.
>
> [9] Jin et al. Simultaneously Learning Stochastic and Adversarial Episodic MDPs with Known Transition. NeurIPS, 20.
>
> [10] Huang, Jiatai, Yan Dai, and Longbo Huang. "Adaptive best-of-both-worlds algorithm for heavy-tailed multi-armed bandits." International Conference on Machine Learning. PMLR, 2022.
>
> [11] Wei, Chen-Yu, and Haipeng Luo. "More adaptive algorithms for adversarial bandits." Conference On Learning Theory. PMLR, 2018.
>
> [12] Cheng, Duo, Xingyu Zhou, and Bo Ji. "Taming Heavy-Tailed Losses in Adversarial Bandits and the Best-of-Both-Worlds Setting." The Thirty-eighth Annual Conference on Neural Information Processing Systems, 2024.
>
> [13] Genalti, Gianmarco, et al. "$(ε, u) $-Adaptive Regret Minimization in Heavy-Tailed Bandits." The Thirty Seventh Annual Conference on Learning Theory. PMLR, 2024.

---

> > ### Author Response · Authors · 2024-12-02
> > **Another Reminder**
> >
> > Dear Reviewer,
> >
> > As the author reviewer discussion period is ending soon, we sincerely thank you for the invaluable feedback. Should our response effectively address your concerns, we kindly hope that you could consider raising the score rating for our work. We will also be happy to address any additional queries or points.

---

### Official Review · Reviewer_jQU5 · 2024-11-04

**Soundness:** 3
**Presentation:** 3
**Contribution:** 3
**Rating:** 8
**Confidence:** 4

**Summary:**

The paper addresses the heavy-tailed MAB problem, a variant of the stochastic bandit problem where rewards are sampled from distributions having potentially infinite variance. The main contribution of this work is to provide an algorithm with tight regret guarantees in both the stochastic and adversarial HTMAB problem. While the performance in the stochastic setting is worse (not in terms of T) than existing algorithms (e.g. AdarUCB), the algorithm simultaneously deals with the two settings and is tight in both the instance-dependent and independent sense.

**Strengths:**

The paper is written, and it provides an exhaustive review of the existing literature.

The contribution is clear, and it is well highlighted which open question the paper addresses.

The paper also presents some nice technical contributions in the algorithm and in the proofs.

**Weaknesses:**

Overall, the contribution is limited when it comes to applications since adversarial HTMABs are uncommon in the real world and the literature. Instead, in the purely stochastic setting, the algorithm does slightly worse than AdaRUCB by a factor of  $\log \frac{\sigma^\alpha}{\Delta_{min}}$ (Genalti et al.)

**Questions:**

- How do applications justify the adversarial HTMAB? Can you please provide some examples?

- I think it would be interesting to highlight the trade-off between ADAR-UCB and UniInf more. How is the best-of-both-worlds property related to the extra factor in the stochastic setting's regret bound? Does your algorithm require extra round-robin turns (as in Adar-UCB)?

- Do you know what the optimal performance would be without the truncated non-negativity assumption? Are there any known lower bounds for the problem without this assumption?

- It would be interesting to understand if alternative (and possibly weaker) assumptions can lead to the same performances (I would like to point out that Theorems 2 and 3 from Genalti et al. don't necessarily imply that this specific assumption is required, but rather that without any assumptions such performances are unattainable).

---

> ### Author Response · Authors · 2024-11-22
> **Author Responses (1/2)**
>
> Thank you very much for your time and effort in reviewing our paper! Please find our responses to your comments below. We will be happy to answer any further questions.
>
>
> ### Weaknesses
>
> > W1: Overall, the contribution is limited when it comes to applications since adversarial HTMABs are uncommon in the real world and the literature
>
> A: It is important to note that heavy-tailed distributions are indeed common in practical scenarios across various sectors. For instance, financial markets [4], medical imaging [5], and online learning algorithms [8] often encounter heavy-tailed loss distributions. Additionally, the relevance of Heavy-tailed Multi-Armed Bandits (HTMABs) extends to real-world applications such as network routing [9] and algorithmic portfolio management [10].
>
> The aspect of designing algorithms that perform well in both stochastic and adversarial settings is crucial for applications such as network scheduling, as evidenced in [6, 7]. Moreover, the increasing interest in the adversarial robustness and Best-of-Both-Worlds (BoBW) properties of HTMAB within the research community [2, 3] further validates our focus. We believe that there exists huge potential in applying efficient learning of HTMABs for both stochastic and adversarial environments in solving some interesting and difficult real world tasks.
>
>
> >W2: In the purely stochastic setting, the algorithm does slightly worse than AdaRUCB by a factor of $\log \frac{\alpha}{\Delta_{\min}}$ (Genalti et al.).
>
> A: Thank you for the observations. It is true that in a purely stochastic setting, our algorithm uniINF shows a slight performance decrement relative to AdaRUCB, specifically by a factor of $\log \frac{\sigma^\alpha}{\Delta_{\min}}$ as detailed in [1]. However, it is crucial to note that [1] is tailored exclusively for stochastic environments, utilizing extensive statistical techniques that enhance arm selection precision which result in the extra pull to the chosen arm, and does not have any performance guarantess in the adversarial setting.
>
> In contrast, our uniINF algorithm, built on the FTRL framework, is designed to handle both stochastic and adversarial environments effectively, differing significantly from [1]'s UCB-based algorithm. This adaptability is particularly valuable in real-world applications where conditions can abruptly shift from stochastic to adversarial, a scenario not addressed by the UCB-based strategies like AdaRUCB.
>
> While we acknowledge the slightly worse regret bounds in stochastic settings, uniINF is pioneering in its capability to deliver BoBW performance in the context of HTMABs. Further eliminating the log factors to match the optimal regret in the stochastic setting is an interesting future direction.
>
> ### Questions
> >Q1: Examples for application of adversarial HTMAB.
>
> A: Please see our answer in W1 above.
>
> >Q2: Trade-off between Adar-UCB.
>
> A: Please see our discussion about the  extra logarithmic term in W2 above.
>
> >Q3: The optimal performance without the truncated non-negativity assumption.
>
> A: We would like to thank the reviewer for the comment.
> To the best of our knowledge, the investigations about the truncated non-negativity loss assumption (Assumption 1) can be summarized by the following table. Moreover, we have modified the statement in our revision (see Lines 187-190) and included a detailed discussion in Appendix B.
>
> | Assumptions | Known $(\sigma,\alpha)$ | Unknown $(\sigma,\alpha)$ |
> |---|---|---|
> |With Assumption 1 | HTINF achieves BoBW [2] | uniINF achieves BoBW (**This paper**) |
> | Weaker than Assumption 1? | SAO-HT achieves BoBW [3] (see below) | `Open` |
> |Without Any Assumption | SAO-HT achieves BoBW [3] | No BoBW possible, Theorems 2 & 3 in [1] |
>
> In particular, as shown in the table,
> - Theorems 2 & 3 in [1] present the lower bound for HTMABs without Assumption 1 and prior knowledge of $(\sigma, \alpha)$, highlight that in parameter-free setups, achieving optimal worst-case regret guarantees is impossible unless further assumptions -- which may or may not be strictly weaker than Assumption 1 -- are made.
> - Our paper demonstrates that, in parameter-free setups, Assumption 1 is sufficient for achieving a BoBW guarantee.
> - Recent work [3] justified that when parameters $(\sigma,\alpha)$ are known, BoBW is achievable without any assumptions (that is, Assumption 1 is redundant when parameters are known).
> - It remains an open question whether a weaker assumption than Assumption 1 could also support BoBW guarantees when $(\sigma, \alpha)$ are unknown.
>
> ---
>
> **We hope our responses fully address your concerns. If so, we wonder if you could kindly consider raising your score rating? Meanwhile, we are also more than happy to answer any further questions. Thank you once again for your review!**

---

> > ### Comment · Reviewer_jQU5 · 2024-11-26
> >
> > Thank you very much for your clarification. After reading your responses and communication with other reviewers, I decided to raise my score, as the paper gives a clear contribution to this literature, which recently has been of interest to researchers.

---

> > > ### Author Response · Authors · 2024-11-26
> > >
> > > Thank you very much for your valuable suggestions and positive feedback!

---

> ### Author Response · Authors · 2024-11-22
> **Author Responses (2/2)**
>
> ### References
>
> [1] Genalti, Gianmarco, et al. "$(ε, u) $-Adaptive Regret Minimization in Heavy-Tailed Bandits." The Thirty Seventh Annual Conference on Learning Theory. PMLR, 2024.
>
> [2] Huang, Jiatai, Yan Dai, and Longbo Huang. "Adaptive best-of-both-worlds algorithm for heavy-tailed multi-armed bandits." International Conference on Machine Learning. PMLR, 2022.
>
> [3] Cheng, Duo, Xingyu Zhou, and Bo Ji. "Taming Heavy-Tailed Losses in Adversarial Bandits and the Best-of-Both-Worlds Setting." The Thirty-eighth Annual Conference on Neural Information Processing Systems, 2024.
>
> [4] Cont, Rama. "Empirical properties of asset returns: stylized facts and statistical issues." Quantitative finance 1.2 (2001): 223.
>
> [5] Hamza, A. Ben, and Hamid Krim. "Image denoising: A nonlinear robust statistical approach." IEEE transactions on signal processing 49.12 (2001): 3045-3054.
>
> [6] Allan Borodin, Jon Kleinberg, Prabhakar Raghavan, Madhu Sudan, and David P. Williamson. 1996. Adversarial queueing theory. In Proceedings of the twenty-eighth annual ACM symposium on Theory of Computing (STOC '96). Association for Computing Machinery, New York, NY, USA, 376–385. https://doi.org/10.1145/237814.237984
>
> [7] Huang, Jiatai, Leana Golubchik, and Longbo Huang. "When Lyapunov Drift Based Queue Scheduling Meets Adversarial Bandit Learning." IEEE/ACM Transactions on Networking (2024).
>
> [8] Zhang, Jingzhao, et al. "Why are adaptive methods good for attention models?." Advances in Neural Information Processing Systems 33 (2020): 15383-15393.
>
> [9] Liebeherr, Jörg, Almut Burchard, and Florin Ciucu. "Delay bounds in communication networks with heavy-tailed and self-similar traffic." IEEE Transactions on Information Theory 58.2 (2012): 1010-1024.
>
> [10] Gagliolo, Matteo, and Jürgen Schmidhuber. "Algorithm portfolio selection as a bandit problem with unbounded losses." Annals of Mathematics and Artificial Intelligence 61 (2011): 49-86.

---

### Official Review · Reviewer_HjBa · 2024-11-08

**Soundness:** 3
**Presentation:** 3
**Contribution:** 3
**Rating:** 8
**Confidence:** 4

**Summary:**

The paper studies Heavy-Tailed MultiArmed Bandits (HTMAB) problem. The main contribution of the paper is to design an optimal algorithm that achieves both Best of-Both-Worlds (BoBW) and Parameter-free properties for HTMAB, where BoBW means that the algorithm performs optimally in both stochastic and adversarial environments and Parameter-free means that the algorithm do not need to know the the heavy-tail parameters in advance.

**Strengths:**

1. The paper is well written. The theoretical results and proof appear to be correct.
2. The paper achieves worst-case BoBW optimal regret for HTMAB, which improves previous results proposed in [Huang 2022].

**Weaknesses:**

1. The paper should include the comparisons with previous scale-free MAB works, e.g. [1-5]. Specifically, the algorithm structure proposed in the paper seems very close to the one proposed in [3], which also uses the clipping/skipping technique and inf regularization. The differences should be further clarified.
2. Assumption 1 is a bit weird. I can understand why it is unavoidable, but I suggest that the authors can give the best (not worst-case optimal) upper bounds we can get without this assumption.

**Questions:**

When $\alpha, \sigma$ known, it is trivial to use regular FTRL based algorithm to achieves (nearly) optimal worst-case regret for adversarial bandit problems with potentially heavy-tailed losses (fix the clipping bound $[-r,r]$ with $r=\sigma T^{1/\alpha}K^{-1/\alpha}$ and use Theorem 4 in [6]). When $\alpha, \sigma$ are unknown, intuitively, it suffices to use the adaptive clipping bound according to the empirical estimation of $\alpha, \sigma$ (Line 6 of ALG 1). Is the high-level idea of the algorithm in this paper the one I described?


References:
[1] Putta, Sudeep Raja, and Shipra Agrawal. "Scale-free adversarial multi armed bandits." International Conference on Algorithmic Learning Theory. PMLR, 2022.

[2] Chen, Mingyu, and Xuezhou Zhang. "Scale-free Adversarial Reinforcement Learning." arXiv preprint arXiv:2403.00930 (2024).

[3] Chen, Mingyu, and Xuezhou Zhang. "Improved Algorithms for Adversarial Bandits with Unbounded Losses." arXiv preprint arXiv:2310.01756 (2023).

[4] Huang, Jiatai, Yan Dai, and Longbo Huang. "Banker online mirror descent: A universal approach for delayed online bandit learning." International Conference on Machine Learning. PMLR, 2023.

[5] Hadiji, Hédi, and Gilles Stoltz. "Adaptation to the range in k-armed bandits." Journal of Machine Learning Research 24.13 (2023): 1-33.

[6] Wei, Chen-Yu, and Haipeng Luo. "More adaptive algorithms for adversarial bandits." Conference On Learning Theory. PMLR, 2018.

---

> ### Author Response · Authors · 2024-11-22
> **Author Responses (1/2)**
>
> Thank you very much for your time and effort in reviewing our paper! Please find our responses to your comments below. We will be happy to answer any further questions you may have.
>
> ### Weaknesses
> > W1: The paper should include the comparisons with previous scale-free MAB works, e.g. [1-5].
>
> A: We appreciate your valuable suggestion. In our revision, we have already enhanced the manuscript by incorporating comparison with previous scale-free MAB works in Appendix A. We highlight the key difference between them and our result below.
>
> Primarily, scale-free MABs results like [1-5] address scenarios where losses are bounded uniformly by some constant $C > 0$, unknown to the algorithm. Their regret guarantees are contingent upon the constant $C$. In contrast, for heavy-tailed MABs, such bounds may be prohibitively large. We thus propose a different approach to avoid the regret bound being dependent on the maximum loss magnitude, but instead, to make it depend only on the heavy-tail parameters $\sigma$ and $\alpha$, thereby offering a more robust solution in environments with heavy-tailed distributions.
>
> >W2: The algorithm structure proposed in the paper seems very close to the one proposed in [3].
>
> A: We appreciate your observation but would like to clarify significant differences between our work and [3]. While [3] addresses scale-free MABs with regret tied to the maximum loss magnitude $\ell_\infty$, our approach targets heavy-tailed MABs and aims for a regret dependent solely on the unknown heavy-tailed parameters $(\sigma, \alpha)$, instead of the maximum loss magnitude which is much larger than $\sigma$. This distinction is crucial because, in scale-free MAB scenarios, algorithms typically endure a total regret of $\tilde O(\ell_\infty \sqrt{KT})$ in the worst case. However, for $T$ i.i.d. $(\sigma, \alpha)$-heavy-tailed loss samples, the expected maximum loss can reach as high as $O(\sigma T^{1/\alpha})$. Therefore, by reducing heavy-tailed MABs to scale-free MABs, exsisting methods can only guarantee a sub-optimal guarantee of $O(T^{1/\alpha + 1/2})$.
>
> Technically, [3]'s algorithm relies only on clipping operations, whereas our algorithm adopts both clipping and skipping (see Line 6 of Algorithm 1) to achieve a parameter-free BoBW guarantee in heavy-tailed MABs. Therefore, our algorithm is substantially different from that of [3].
>
> >W3: Assumption 1 is a bit weird.
>
> A:  We would like to thank the reviewer for the comment. As pointed out by the reviewer, our previous discussion statement might cause confusion. Thus we have modified the statement in our revision (see Lines 187-190) and included a detailed discussion in Appendix B.
>
> It is important to note that the truncated non-negativity loss assumption (Assumption 1) is not unique to our study. In fact, it was also used in references [4] and [7], and is indeed a relaxation of the more common "non-negative losses" assumption found in the MAB literature. For a better comparison, we summarize the exisiting results for the relationship between heavy-tailed MABs and Assumption 1 in the following table.
>
> | Assumptions | Known $(\sigma,\alpha)$ | Unknown $(\sigma,\alpha)$ |
> |---|---|---|
> |With Assumption 1 | HTINF achieves BoBW [9] | uniINF achieves BoBW (**This paper**) |
> | Weaker than Assumption 1? | SAO-HT achieves BoBW [8] (see below) | `Open` |
> |Without Any Assumption | SAO-HT achieves BoBW [8] | No BoBW possible, Theorems 2 & 3 in [7] |
>
> In particular, as shown in the table,
> - Theorems 2 & 3 in [7] highlight that in parameter-free setups, achieving optimal worst-case regret guarantees is impossible unless further assumptions -- which may or may not be strictly weaker than Assumption 1 -- are made.
> - Our paper demonstrates that, in parameter-free setups, Assumption 1 is sufficient for achieving a BoBW guarantee.
> - Recent work [8] justified that when parameters $(\sigma,\alpha)$ are known, BoBW is achievable without any assumptions (that is, Assumption 1 is redundant when parameters are known).
> - It remains an open question whether a weaker assumption than Assumption 1 could also support BoBW guarantees when $(\sigma, \alpha)$ are unknown.

---

> ### Author Response · Authors · 2024-11-22
> **Author Responses (2/2)**
>
> ### Questions
>
> A: Thank you for sharing your intuition! We are sorry but we are not sure whether we fully understand it, so we try to discuss the potential difficulties one might encounter following your idea below.
>
> Suppose one uses a fixed clipping threshold $r = \sigma T^{1/\alpha}K^{-1/\alpha}$ and then run an MAB algorithm for $[-1,1]$-bounded losses via rescaling the clipped losses after clipping by $1/r$. Then, even if there is no error introduced by this clipping operation, the regret would be $\tilde O(r\cdot \sqrt{KT})$ in the worst case. It is unclear how Theorem 4 in [6] can help obtain a total regret bound much tighter than $\tilde O(\sqrt{KT})$ -- to do so, one essentially needs to control the sample-path loss variances $Q_{t,i}$'s, which is highly non-trivial. Also, we are not sure how we can derive an "empirical estimation of $\alpha, \sigma$" via Line 6 of Algorithm 1. Our Line 6 performs clipping and skipping operations but does not estimate $\alpha$ or $\sigma$. In fact, due to the adversarial nature of our setup, we believe such estimators are highly challenging to construct as well.
>
> ---
>
> **We hope our responses fully address your concerns. If so, we wonder if you could kindly consider raising your score rating? Meanwhile, we are also more than happy to answer any further questions. Thank you once again for your review!**
>
> ### References
> [1] Putta, Sudeep Raja, and Shipra Agrawal. "Scale-free adversarial multi armed bandits." International Conference on Algorithmic Learning Theory. PMLR, 2022.
>
> [2] Chen, Mingyu, and Xuezhou Zhang. "Scale-free Adversarial Reinforcement Learning." arXiv preprint arXiv:2403.00930 (2024).
>
> [3] Chen, Mingyu, and Xuezhou Zhang. "Improved Algorithms for Adversarial Bandits with Unbounded Losses." arXiv preprint arXiv:2310.01756 (2023).
>
> [4] Huang, Jiatai, Yan Dai, and Longbo Huang. "Banker online mirror descent: A universal approach for delayed online bandit learning." International Conference on Machine Learning. PMLR, 2023.
>
> [5] Hadiji, Hédi, and Gilles Stoltz. "Adaptation to the range in k-armed bandits." Journal of Machine Learning Research 24.13 (2023): 1-33.
>
> [6] Wei, Chen-Yu, and Haipeng Luo. "More adaptive algorithms for adversarial bandits." Conference On Learning Theory. PMLR, 2018.
>
> [7] Genalti, Gianmarco, et al. "$(ε, u) $-Adaptive Regret Minimization in Heavy-Tailed Bandits." The Thirty Seventh Annual Conference on Learning Theory. PMLR, 2024.
>
> [8] Cheng, Duo, Xingyu Zhou, and Bo Ji. "Taming Heavy-Tailed Losses in Adversarial Bandits and the Best-of-Both-Worlds Setting." The Thirty-eighth Annual Conference on Neural Information Processing Systems, 2024.
>
> [9] Huang, Jiatai, Yan Dai, and Longbo Huang. "Adaptive best-of-both-worlds algorithm for heavy-tailed multi-armed bandits." International Conference on Machine Learning. PMLR, 2022.

---

> ### Comment · Reviewer_HjBa · 2024-11-26
>
> Thanks for the detailed responses! Most of my concerns are addressed. I raised my score now.

---

> > ### Author Response · Authors · 2024-11-26
> >
> > Thank you very much for your valuable suggestions and positive feedback!

---

### Author Response · Authors · 2024-11-22

We sincerely thank all reviewers for their efforts in reviewing our paper and for their valuable comments. We have revised our manuscript based on feedback and submitted the revision. Changes in the revision are highlighted in pink and detailed below:

1. We have revised the Related Works section and added a comprehensive literature review in Appendix A, which addresses the reviewers' comments. This includes enhanced comparisons of scale-free MABs and further discussion on data-dependent learning rates.

2. We thank the reviewer for pointing out potential confusion in our initial discussion of the truncated non-negative assumption (Assumption 1). We have modified the statement (see Lines 187-190) and included a detailed discussion in Appendix B to clarify this assumption further.

3. We have refined our description of the novel analysis for multiplicative stability in heavy-tailed scenarios, detailed in Lines 420-424. This revision aims to more clearly introduce our innovative approach and its implications.

---

### Meta-Review · Area_Chair_ZiKp · 2024-12-21

**Metareview:**

This paper proposes a best-of-both-worlds algorithm for the heavy-tailed multi-armed bandit problem, which achieves optimal performance in both stochastic and adversarial environments. A significant strength of the proposed algorithm is its adaptability to the (unknown) heavy-tail parameter. However, its limitations include the regret upper bound in the stochastic setting depending solely on the minimum gap $\Delta_{\min}$ rather than the individual gaps $(\Delta_i)$, and the requirement of the truncated-non-negativity assumption (Assumption 1). Regarding the latter point, the authors have appropriately justified its necessity by referencing the lower bound established in prior work [Genalti et al., 2024]. The authors have adequately addressed the reviewers' concerns and questions. Given the consensus among the reviewers to accept the paper, I support its acceptance.

Additionally, there are areas for improvement as follows:
* The current text in Lines 213-253 may give the impression that this paper is the first to demonstrate the best-of-both-worlds property using log-barrier regularization in FTRL. However, previous studies, such as (Wei & Luo, 2018) and (Ito, 2021b), have also shown that the best-of-both-worlds property can be achieved using log-barrier regularization. It would be better to revise the description to avoid potential misunderstandings.
> While log-barrier regularizers were commonly used in the literature for data-adaptive bounds such as small-loss bounds (Foster et al., 2016), path-length bounds (Wei & Luo, 2018), and second-order bounds (Ito, 2021b), this paper introduces novel analysis illustrating that log-barrier regularizers also provide environment-adaptivity for both stochastic and adversarial settings.
* Regarding Assumption 2, while it is true that this assumption is common in prior work, it may also be worth mentioning that Tsallis-INF (Zimmert & Seldin, 2019) and its extensions do not require this assumption; The analysis of Tsallis-INF is provided in (Ito, 2021b), and its extension is presented in (Jin et al., 2023).

**Additional Comments On Reviewer Discussion:**

The reviewers raised concerns regarding comparisons with prior work, technical novelty, and the strength of the assumptions. However, the authors have appropriately addressed these concerns through their responses and revisions to the paper.

---

### Decision · Program_Chairs · 2025-01-22

Accept (Spotlight)